

# How to represent human behavior and decision making in Earth system models? A guide to techniques and approaches

Finn Müller-Hansen[1,2], Maja Schlüter[3], Michael Mäs[4], Rainer Hegselmann[5,6], Jonathan F. Donges[1,3], Jakob J. Kolb[1,2], Kirsten Thonicke[1], and Jobst Heitzig[1]

[1]Potsdam Institute for Climate Impact Research, Telegrafenberg A31, 14473 Potsdam, Germany
[2]Department of Physics, Humboldt University Berlin, Newtonstraße 15, 12489 Berlin, Germany
[3]Stockholm Resilience Center, Stockholm University, Kräftriket 2B, 114 19 Stockholm, Sweden
[4]Department of Sociology and ICS, University of Groningen, Grote Rozenstraat 31, 9712 TG Groningen, The Netherlands
[5]Frankfurt School of Finance & Manangement, Sonnemannstraße 9-11 60314 Frankfurt am Main, Germany
[6]Bayreuth Research Center for Modeling and Simulation, Bayreuth University, Universitätsstrasse 30, 95440 Bayreuth, Germany

*Correspondence to:* Finn Müller-Hansen (mhansen@pik-potsdam.de)

**Abstract.** In the Anthropocene, humans have a critical impact on the Earth system and vice versa, which can generate complex feedback processes between social and ecological dynamics. Integrating human behavior into formal Earth System Models (ESMs), however, requires crucial modeling assumptions about actors and their goals, behavioral options and decision rules, as well as modeling decisions regarding human social interactions and the aggregation of individuals' behavior. In this tutorial

review, we compare existing modeling approaches and techniques from different disciplines and schools of thought dealing with human behavior at various levels of decision making. Providing an overview over social-scientific modeling approaches, we demonstrate modelers' often vast degrees of freedom but also seek to make modelers aware of the often crucial consequences of seemingly innocent modeling assumptions.

After discussing which socio-economic units are potentially important for ESMs, we review models of individual decision

making that correspond to alternative behavioral theories and that make diverse modeling assumptions about individuals' preferences, beliefs, decision rules, and foresight. We discuss approaches to model social interaction, covering game theoretic frameworks, models of social influence and network models. Finally, we elaborate approaches to study how the behavior of individuals, groups and organizations can aggregate to complex collective phenomena, discussing agent-based, statistical and representative-agent modeling and economic macro-dynamics. We illustrate the main ingredients of modeling techniques with

examples from land-use dynamics as one of the main drivers of environmental change bridging local to global scales.

## 1 Introduction

Even though Earth system models (ESMs) are used to study and project the human impact on the complex interdependencies between various compartments of the Earth, humans are not represented in these models. ESMs consider the human influence usually in terms of scenarios, comparing alternative narratives about the future development of key socio-economic character-

istics of human societies. For instance, the IPCC scenario approach uses economic integrated assessment models to compute



plausible future emission pathways for energy and land use (RCPs, SSPs). These emission projections are then used as external input in Earth system models to study changes in climate and the consequent natural impacts (Moss et al., 2010; IPCC, 2014). These natural impacts may be translated back to socio-economic impacts and fed again into the scenario process, leading to an iterative process. However, the dynamic and potentially complex interplay of dynamics of the natural Earth system and human

social, cultural and economic responses are not captured.

In the proclaimed Anthropocene epoch, human societies are a dominant geological force interfering with biophysical Earth system processes at all relevant scales (Crutzen, 2002; Maslin and Lewis, 2015). However, changing environmental conditions also alter human behavior. For example, climate change will affect how humans use land and consume energy. Likewise, perceived environmental risks modify consumption and mobility patterns. Therefore, with increasing human impact on the

Earth system, feedbacks between shifts in the biophysical Earth system and human responses will gain importance (Palmer and Smith, 2014; Verburg et al., 2016; Donges et al., in press). To get an overview over possible feedback mechanisms, Donges et al. (in prep. for this special issue) identify interactions between the social, metabolic and environmental spheres of the Earth system including humans.

Incorporating human behavior in ESMs is a complex endeavor. Modeling the interaction between various nonlinear compo-

nents of the Earth system is already a huge challenge, even though traditional ESMs rely on precise natural laws. In order to capture feedbacks between biophysical and social dynamics, it is necessary to explicitly model human decision making and behavior, which can be very heterogeneous. Accordingly, scientific understanding of the determinants of individuals' behavior as well as its collective consequences is still limited. Furthermore, human action is influenced by contingent and socially formed norms and institutions. This allows a view on social systems as socially constructed realities, which is in stark contrast

to the positivist epistemology of one objective reality prevalent in the natural sciences. These epistemological questions may cause misunderstandings between natural and social scientists and show to the fundamental difference between rules in social systems and natural laws.

In the past decades, technological advances and the Internet have brought about unprecedented amounts of data about individual behavior and have led to a rapid growth in computational power. With these advances, new models that include

human decision making and behavior could move beyond current approaches and describe for example changes in social norms and preferences, consumer behavior and/or social structure besides purely economic relationships. Contrary to conventional approaches, such coupled models would allow exploring possible complex nonlinear dynamics in the Earth system and reveal potential social-environmental tipping points and regime shifts (Filatova et al., 2016).

Here, we provide a guide for Earth system modelers to existing modeling approaches describing human behavior and deci-

sion making. Following Weber (1978), we define decision making as the cognitive process of choosing consciously between alternative actions. Actions are intentional and subjectively meaningful activities of an agent. Behavior, in contrast, is a broader concept that also includes unconscious and reflexive activities, such as habits. In Earth system models, only those human decisions and responses are relevant that have considerable impacts on the Earth system. They result from behavior of a large number of individuals or amplified decisions, e.g. through the social position of the decision-maker or technology. Therefore,

this paper also cover techniques to model individuals' interactions and to aggregate individual's behavior and interactions to a





macro-level. On the micro-level, relevant decisions include for instance reproduction, consumption and production of energy- and material-intensive products, place of living and land use. These decisions lead to aggregate and long-term dynamics in population, production and consumption patterns and migration between countries as well as urban and rural areas.

The relation between individual agents and social collectives and structures has been the reason for considerable debate

in the social sciences: In the social-scientific tradition of methodological individualism[1], the analysis aims to explain social macro-phenomena, e.g., phenomena at the level of social collectives such as groups, organizations, and societies, with theories of individual behavior (Coleman, 1994; Udehn, 2002; Homans, 1951). This approach deviates from structuralist traditions, which claim that collective phenomena are of their own kind and can, thus, not be traced back to the behavior of individuals (Durkheim, 2014). Positions between these two extremes emphasize the interdependency of individual agents and social

structure. Structure is understood as a phenomenon emerging from the interactions between agents and stabilizes particular behaviors (Giddens, 1984). While it very much depends on the purpose of the given modeling exercise whether the model should represent individuals or collectives (e.g. households, neighborhoods, cities, countries), we mainly focus here on a research tradition that acknowledges that complex and unexpected collective phenomena can arise from the interplay of individual behavior.

There are diverse social science theories explaining human behavior and decision making in environmental and ecological contexts, for example in the fields of environmental and ecological economics (Perman et al., 2003; van den Bergh, 2001), environmental sociology and psychology (Pellow and Nyseth Brehm, 2013), and many others. In this paper, we focus on mathematical and computational models of human decision making and behavior. Here, we understand the terms 'modeling approach' and 'modeling technique' as a class of mathematical or computational structures that can be interpreted as a simpli-

fied representation of physical objects and actors or collections thereof, events and processes, causal relations or information flows. Modeling approaches may differ for instance according to (i) variables and parameters that they use to describe the entities of the modeled system, (ii) the logical or functional relationships between modeled entities, (iii) the representation of space and time, if any, and (iv) the kinds of mathematical and computational solution techniques applied to find a solution of the model. The modeling approaches that we review often draw upon theories of human behavior that make – often contested

– assumptions about the structure of decision processes and the resulting behavior. Furthermore, we want to point out that models of social systems can have different purposes, which is important for the choice of modeling approach. The purpose can be either descriptive (helping to answer empirical questions, e.g., which components can explain the system's dynamics) or normative (helping to answer ethical questions, e.g., how should we act or which policy should we choose to reach a certain goal).

Recent reviews focus on existing modeling approaches and theories that are applied in the context of environmental change: For example, Verburg et al. (2016) assess existing modeling approaches and identify challenges for improving these models in order to better understand the Anthropocene. Meyfroidt (2013) and Schlüter et al. (2017) focus on cognitive and behavioral theories in ecological contexts, providing an overview for developers of agent-based land-use and social-ecological models.

---

[1]We note, though, that there are different accounts of methodological individualism and it often remains unclear to what extend structural and interactionist elements can be part of an explanation, see Hodgson (2007).



Cooke et al. (2009) also classify micro- and macro-approaches and review their applications in agro-ecology. The present paper complements this literature, reviewing modeling approaches of individual agent behavior, agent interactions and aggregation. The combination of these three different categories is crucial to describe human behavior at a level relevant for ESMs. Furthermore, this review highlight connections between modeling techniques and their underlying assumptions about human behavior

and discuss criteria to guide modeling choices. The presented composition and classification of approaches into categories was guided by an iterative process that aims at an interdisciplinary understanding.

This paper works with land-use change as a guiding and illustrative example. Land use and land-cover change is the second largest source of greenhouse gases – besides the burning of fossil fuels – and thus contributes strongly to climate change, one of the most challenging environmental problems of our time. Behavioral responses in the land-based sector will play a

crucial role for successful mitigation and adaptation to projected climatic changes, challenging modelers to represent decision making in models of land-use change (Brown et al., 2017). The complexity of land-use change provides various examples how collective and individual decision making interacts with the environment across spatial scales and organizational levels. Land-use models consider environmental conditions as important factors in decision-making processes, giving rise to feedbacks between environmental and socio-economic dynamics (Brown et al., 2016). Furthermore, there are first attempts to integrate

diverse human decision making explicitly into global models by the use of agent functional types in the context of land-use science (Arneth et al., 2014). However, this paper does not provide an exhaustive overview over existing land-use models. For this purpose, the reader is referred to the various reviews in the literature (e.g., Baker, 1989; Brown et al., 2004; Michetti, 2012; Groeneveld et al., 2017).

The rest of the paper is organized as follows. In Section 2, we give an overview over different levels of description of social

systems, the socio-economic units or agents associated with them and the research communities that study them. Sections 3–5 form the main part of the paper. There, we review the different modeling techniques and their underlying assumptions about human decision making and behavior in detail, following a simple tripartition: First, Section 3 introduces approaches to model individual decisions and behavior. Second, Section 4 puts the focus on techniques for modeling interactions between agents. Third, Section 5 reviews different aggregation techniques that allow describing human activities at the system level. Examples

from the land-use context are used throughout these sections to illustrate the modeling techniques in a relevant context. Section 6 provides a discussion of criteria and questions for guiding model selection and important distinctions between models of natural and social systems. The paper concludes with remarks on the many remaining challenges for incorporating human behavior into Earth system models.

## 2   The challenge: Modeling decision making and behavior across different levels of organization

Decision making and behavior of humans can be described and analyzed at different levels of social systems. While decisions are made and behavior is performed by individual humans, it is often useful to not represent individual humans in a model but to treat social collectives, such as households, neighborhoods, cities, organizations and states, as decision makers or agents. However, we argue below that independent of the level of analysis the following main questions are useful to guide the modeling



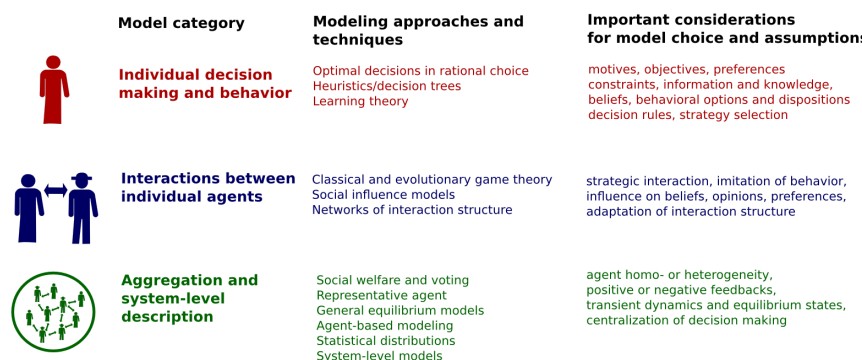

**Figure 1.** Overview of modeling categories, corresponding modeling approaches and techniques discussed in this paper and important considerations for model choice and assumptions about human behavior and decision making.

choices regarding decision making of agents: Which goals do individual agents follow? Which constraints restrict the pursuit of these goals? And finally, according to which decision rules do the agents choose an action? Furthermore, when thinking about how to integrate human decision making into Earth system models, we are generally interested in the outcome of aggregate and collective behavior, i.e. the group outcome of mutually interdependent individual decisions, possibly leading to a joint decision.

Therefore, a considerable part of this paper will be devoted to providing guidelines to modeling approaches that are organized around two additional questions: In which way do individual agents interact? How are individual decisions and interactions aggregated to phenomena at the level of social collectives? Figure 1 gives an overview of the modeling approaches that we introduce in detail in Sections 3–5 and important considerations for model choice and assumptions about human behavior and decision making.

A central challenge for integrating human decision making into Earth system models is the bridging of several levels of social organization and collectives as well as across spatial and temporal scales. Figure 2 shows a hierarchy of socio-economic units, i.e., groups, organizations and structures of individuals that play a crucial role in human interactions with the Earth system. We consider a broad scheme of levels ranging from the individual and micro-level across intermediate levels to the macro- and global level. This hierarchy of socio-economic units is not only distinguishable by level of complexity but also by the different

spatial scales involved: For instance, some individuals may also operate at the global level while transnational organizations may also have impacts on the local level. Because some socio-economic units span various scales there is no canonical mapping. Especially in the context of human-environment interactions in Earth system models, scaling and spatial extent are important issues (Gibson et al., 2000). Furthermore, we note that the clear distinction between a micro- and macro-level may result in neglecting intermediate levels and treating very different phenomena alike. For instance, many economic models describe

both small businesses and transnational corporations as actors on the micro-level and model their decision processes with the same set of assumptions, even though they operate very differently. One of the major difficulties for modeling humans in the Earth system is therefore to bridge these diverse levels between individuals and the global population with all its structural complexities.



In Table 1, we summarize the socio-economic units found in Fig 2 and connect them to traditional scientific disciplines and fields that focus on them. Additionally, we name some common theories, frameworks and assumptions that are made about decision making and human behavior.

At the micro-level, models consider individuals, households, families and small businesses. Individuals can make decisions as policy makers, investors, business managers, consumers, resource users, or in various other contexts. Communities and disciplines focusing on this level are the cognitive and behavioral sciences, and related fields. More specifically in the context of human-nature interactions interdisciplinary fields like natural resource management, resource and institutional economics, social-ecological and land systems research. At this level, decisions about lifestyle, consumption, individual natural resource use, migration and reproduction are particularly relevant in the environmental context. Individual decisions have to be taken by a large number of individuals or have to be multiplied by organizations, institutions or technology to become relevant at the level of the Earth system. Participation in collective decision processes, such as voting, also has potential consequences at higher levels.

At various intermediate levels, communities and organizations like firms, political parties, labor unions, educational institutions, non-governmental and lobby organizations play a crucial role in shaping national economic and policy decisions and therefore have a huge impact on aggregate behavior. Governments at different levels and representing different territories, from cities to nation states, enact laws that strongly frame the condition for economic and social activities of their citizens. Fields that are concerned with this level include sociology, political science, economics, management science and anthropology. Important decisions for the Earth system context include environmental regulations and standards, production and distribution of commodities and assets, trade, extraction and use of natural resources (e.g., mining, forestry, burning of fossil fuels) and the development and building of physical infrastructures (e.g., roads, dams, power and telecommunication networks).

At the global level, multinational companies and intergovernmental organizations negotiate decisions. This level may be remote from most individuals, but it has nevertheless huge impacts on policy and business decisions. Often this level provides framing for activities on lower organizational levels and thus strongly influences the problem statements and perceived solutions for instance regarding environmental issues. Disciplines that focus on this level include macroeconomics, international relations, as well as most of the disciplines mentioned in the previous section. Decisions especially influencing the Earth system at this level are for instance international climate and trade agreements, decisions of internationally operating corporations and financial institutions, and the adoption of global frameworks like the UN Sustainable Development Goals (United Nations General Assembly, 2015).

An overarching question that has triggered considerable debate between different disciplines is the allocation of agency at different levels of description. Even if individuals can decide between numerous options, the perception of options and decisions between them are shaped by social context and institutional embedding. Institutions[2] and organizations can display their own dynamics and lead to outcomes unintended by the individuals. On the other hand, there can be disruptive changes

---

[2]The notion of institution is used in the literature with slightly different meanings: (1) formal and informal rules that shape behavior, (2) informal social order, i.e. regular patterns of behavior, and (3) organizations. Here, we adopt an understanding of institutions as formal (e.g., law, property rights) or informal rules (e.g., norms, religion). However, formal rules often manifest in social, political and economic organizations and informal rules may be shaped by them.



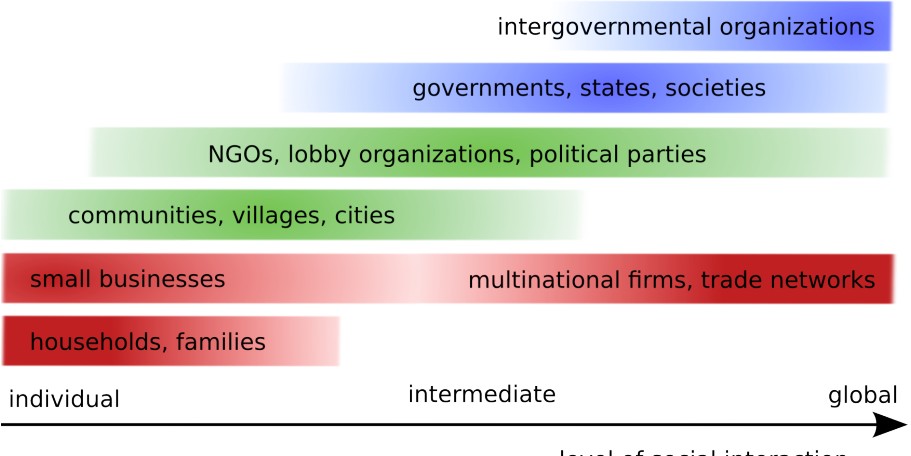

**Figure 2.** Socio-economic units and their corresponding level and scales.

in institutional development brought on by social movements. This is important to bear in mind when choosing a level of description for modeling a given problem.

In the following three sections, we introduce the modeling techniques that are used in the literature to describe human behavior, interactions between individuals, and to aggregate them between the different levels. We start this overview at the level of individual behavior.

## 3   Modeling individual behavior and decision making

In a nutshell, models of individual decision making and behavior differ with regard to their assumptions about three crucial determinants of human choices (Hedstrom, 2005; Lindenberg, 2001, 1990, 1985). First, all models assume that individuals have motives or goals. That is, agents rank goods or outcomes in terms of their desirability and seek to realize highly ranked outcomes. For instance, learning theories assume that actors evaluate the outcome of their choices and that satisficing decisions are reinforced. Other models, such as rational choice theory, make more complex assumptions about preference relations (von Neumann and Morgenstern, 1944). Another prominent but debated assumption is that motives are assumed to be stable over time. Stable preferences are included to prevent researchers from developing trivial explanations, as a theory that models a given change in behavior only based on changed motives does not have explanatory power. However, empirical research shows that preferences can change even in relatively short time frames (Ackermann et al., 2016). Furthermore, changing individuals' preferences is an important way to affect their behavior, making flexible preferences particularly interesting for Earth System modelers.

Second, all models make assumptions about restrictions and opportunities that constrain or help the agents to follow the motives or goals. For instance, each behavioral option comes with certain costs (e.g., money and time) and decision makers



**Table 1.** Overview of particular levels of description of socio-economic units, associated scientific fields/communities and some common approaches and assumptions about decisions and behavior. The list gives a broad overview but is far from being exhaustive.

| Level | Socio-economic unit | Field/Community | Common approaches and theories | Common assumptions about decision making |
|---|---|---|---|---|
| Micro | Individual humans | Psychology, neuroscience, sociology, economics, anthropology | Rational choice, bounded rationality, heuristics, learning theory, complex cognitive architectures | [All assumptions presented in this collumn] |
| | Households, families, small businesses | Economics, anthropology | Rational choice, heuristics, social influence | Maximization of consumption, leisure, profits |
| Intermediate | Communities (villages, neighborhoods), cities | Sociology, anthropology, urban studies | Social influence, networks | Transmission and evolution of cultural traits and traditions |
| | Political parties, NGOs, lobby organizations, educational institutions | Political science, sociology | Social influence on networks, strategic decision making, public/social choice, evolutionary interactions | Influenced by beliefs and opinions of others, agents form coalitions (cooperate) to achieve goals |
| | Governments | Political science, operations research | Strategic decision making, cost-benefit and welfare analysis, multi-criteria decision making | Agents choose for the common good |
| | Nation states, societies | Economics, political science, sociology | welfare maximization, social choice | Majority vote |
| Global | Multinational firms, trade networks | Economics, management science | Rational choice | Maximization of profits or shareholder value |
| | Intergovernmental organizations | Political science (international relations) | Strategic decision making, cost-benefit analysis | Coalition formation |

form more or less accurate beliefs about these costs and how likely they are to occur, depending on the information available to the agent.

Third, actors apply some decision rule that translates their preferences and restrictions into a choice. Although decision rules differ very much in their complexity, they can be categorized into three types. First, there are decision rules that are forward looking. Rational choice theory, for instance, assumes that individuals list all positive and negative future consequences of a decision and choose the optimal option. Alternatively, backwards looking approaches, such as classical reinforcement learning,





assume that actors remember the satisfaction experienced when they chose a given behavior in the past and choose the behavior that felt best. Finally, there are sideward-looking decision rules, which assume that actors adopt the behavior of others, for instance because they imitate successful others (Kandori et al., 1993). Decision rules are interlinked with assumptions about the agent's cognitive capabilities.

In the remainder of this section, we describe in more detail three important models of individual decision making: models of rational choice, models of bounded rationality and learning models. For each model we discuss typical assumptions about motives, restrictions and decision rules. In Section 6 we provide general guidelines for the choice of model assumptions.

### 3.1    Optimal decisions and utility theory in rational choice models

Rational choice theory is an approach to model goal-oriented decision making. Rational choice models assume that agents

have *preferences*, representing goals that they try to pursue given a number of external *constraints*. Agents choose the action that brings about the most preferred outcome. Some versions of rational choice theory also take into account that agents form *beliefs* about external constraints on their decision options (see beliefs, preferences, constraints (BPC) model, Gintis, 2009). Beliefs are subjective priors that can be shared among agents, but contrary to external constraints they can be wrong. Rational choice theory is a standard model in various social sciences, especially in economics, and has been widely studied also in

mathematics.

The qualification of a decision or action as being rational is subject to ongoing debates. For example, Opp (1999) distinguishes between a strong and weak version of rational choice theory. While the strong version (often referred to as homo economicus) describes purely self-interested agents that have full control and knowledge of their possible actions, information about the probabilities of possible consequences and unlimited capacities to compute the optimal decision to take, a weaker

version relaxes these assumptions. Other authors like Rabin (2002) further distinguish between standard and non-standard assumptions regarding preferences, beliefs and decision-making rules. In the remainder of this subsection, we discuss the different assumptions regarding preferences and beliefs, while in the next subsection (3.2) we introduce decision-making rules that deviate from the standard that agents always choose the optimal action. Usually individual preferences are assumed to be fixed over the relevant time scales, to regard possible outcomes of actions and their personal consequences for the agent

(self-interest or even "selfishness", in particularly assumed by economics scholars), and to take into account risk in some way (see below). In general, however, preferences are completely neutral with regard to their content and, for example, can also concern features of collective decision processes (procedural preferences, e.g., Hansson, 1996; Fehr and Schmidt, 1999) and consequences for others (other-regarding preferences and altruism, e.g., Mueller, 2003; Fehr and Fischbacher, 2003).

In the broadest version, preferences are modeled as *preference relations*, e.g. $x\ P_i\ y$ denoting that individual $i$ prefers $x$

to $y$, where $x$ and $y$ represent outcomes, consequences, processes, combinations thereof, or probability distributions of such. Standard versions of rational choice theory assume that the binary relations $P_i$ are complete (for every pair $(x, y)$ either $xP_iy$ or $yP_ix$) and transitive (if $xP_iy$ and $yP_iz$ then $xP_iz$), although more general preference relations are possible (e.g., Fishburn, 1968; Heitzig and Simmons, 2012). These properties allow it to be represented by a *utility function* $u_i$ with $u_i(x) > u_i(y)$ if





and only if $x\ P_i\ y$.[3]. Utility functions thus specify how combinations of behavioral outcomes satisfy the preferences of the decision makers.

In the context of land use, $i$ could be a farmer and $x$ might denote a state of affairs, where $i$ grows some traditional crops generating a moderate profit. In addition, $y$ could denote an alternative state of affairs where $i$ instead grows some genetically modified hybrid seeds generating more profit but putting $i$ into a strong dependency on the seed supplier. Then, $x\ P_i\ y$ would denote $i$'s preference of $x$ over $y$ because he considers independence valuable enough to make up for the lower profit.

Utility functions are particularly useful in the context of *decision making under uncertainty*[4]. To determine the optimal decision under probabilistic uncertainty, the standard *expected utility theory* is usually applied to calculate the utility $u_i(p)$ of a *lottery* or *risky prospect* (i.e. a probabilistic outcome) $p$ represented by probabilities $p(x)$ as the linear combination $u_i(p) = \sum_x p(x) u_i(x)$. Empirical research however shows that only a minority of people evaluate lotteries in this *risk-neutral* way (Kahneman and Tversky, 1979). The vast majority, however, overestimates small probabilities and shows *risk-aversion* or *risk-seeking* with respect to losses or gains in comparison with expected utility theory. Such decisions are described by *prospect theory*, using the non-linear formula $u_i(p) = \sum_x w(p(x)) v(u_i(x))$ with suitable functions $v$ and $w$ (Kahneman and Tversky, 1979), or by the slightly more complex *cumulative prospect theory* (e.g., Bruhin et al., 2010).

A conceptual example from the land-use context illustrates decision making under risk: A farmer $i$ might face the choice whether to stick to her current crop ($x$) or switch to a different crop ($y$). She may think that with 20% probability the switch will turn out badly, resulting in only a quarter as much yield as with $x$, while with 80% probability, the yield would double. If her utility depends logarithmically on yield and she evaluates this uncertain prospect as described by expected utility theory, her gain from switching to $y$ would be positive. If, however, she is averse to losses and thus conforms to prospect theory, she might evaluate the switch as negative and prefer to stick to $x$.

If behavior and its consequences involve several time points $t$, then *time preferences* and *patience* are often taken into account via *discounting*. Discounted utility quantifies the present desirability of some utility obtained in the future. Therefore, discounting can be used to measure the utility that an individual derives at a given point in time from future consequences of her current decisions. Exponential discounting is often used in models because it is mathematically convenient and *time-consistent*, meaning that it makes no difference at which point in time the evaluation is made. However, empirical research finds that people seem to discount hyperbolically, meaning that their valuation in the short-term declines much faster than in the long-term. This is *time-inconsistent* because people might prefer getting one dollar today over two dollars tomorrow but two dollars in a month and a day over one dollar in a month (Ainslie and Haslam, 1992; Jamison and Jamison, 2011).[5]

---

[3]The utility function $u_i$ is only defined up to positive linear (affine) transformations.

[4]We note that some authors make the distinction between risk as unknown events with measurable probabilities ("known unknowns") as opposed to (fundamental) uncertainty as such events without any knowledge about their probabilities ("unknown unknowns", cp. Knight, 2006). Although fundamental uncertainty may be important in human decision making, we only consider risk here because some forms of fundamental uncertainty cannot be represented in models.

[5]For exponential discounting, future (expected) utility is depreciated with an exponentially decaying factor $u_i(x) = \sum_t \exp(-rt) u_i(x,t)$, while for hyperbolic discounting the factor decays slower in the far future $u_i(x) = \sum_t u_i(x,t)/(1+rt)^s$.





Consider as an example from the land-use context a farmer $i$ who compares different crops not only by next year's expected profit $u_i(x,1)$ but, due to the various crops' different effects on future soil quality, also by future years' profits $u_i(x,t)$ for $t > 1$. Crop $y$ might promise higher yields than $x$ in the short run but lower ones in the long run due to faster soil depletion, so that although $u_i(x,1) > u_i(y,1)$, it might still be that her evaluation of this utility stream is $u_i(x) < u_i(y)$, but only if $i$ is
"patient" enough, i.e., if the discounting rate $r$ is small enough.

In addition, preference aggregation can also be necessary across independent or coupled decisions dealing with several interrelated issues or types of consequences. For example, in the modeling of preferences over *consumption bundles* in consumer theory (Varian, 2010), the utility derived from consuming $n$ apples, $u_{i,a}(n)$ and $m$ pears, $u_{i,p}(m)$, may be combined into a total consumption utility by means of an *additively separable* utility function $u_i(n,m) = u_{i,a}(n) + u_{i,p}(m)$, a
*Cobb-Douglas* utility function $u_i(n,m) = u_{i,a}(n)^\alpha u_{i,p}(m)^{1-\alpha}$, or a *constant elasticity of substitution* (CES) utility function $(u_{i,a}(n)^r + u_{i,p}(m)^r)^{1/r}$ representing different forms of *substitutability of goods*. In the land-use context, farmers' utility from leisure time $h_x$ and consumption enabled by work in the field that increases crop yield $y_x$ might be combined in a similar way (e.g. via a Cobb-Douglas utility function $u_i(x) = y_x^\alpha (12 - h_x)^{1-\alpha}$).

Given constraints by the environment, the available information and the evaluation by utility resulting from each possible
action, rational choice theory assumes that the agent chooses the action with the maximal utility. In models, the resulting optimization problem is solved using tools such as *mathematical programming* (e.g. linear programming) or *calculus of variations* (see e.g., Kamien and Schwartz, 2012; Chong and Zak, 2013). Optimal decisions under constraints are not only discussed as a description of human behavior, but are often taken as the normative benchmark for comparison with other non-optimal approaches that we discuss in the following section.

Regarding decision-modeling in Earth system models, rational choice theory is useful for contexts in which the agents' goals are sufficiently clear, agents can be assumed to posses enough information, time and cognitive resources to assess all available options for action. For instance, individuals' decisions regarding long-term investments or decisions of organizations such as firms or governments in competitive situations can often be assumed to follow a rational action model. However, rational choice can also be a useful assumption when actors make the same decision many times and get immediate feedback, so that
they learn to choose the optimal option. Thus, they behave "as if" they were rational decision makers.

### 3.2   Bounded rationality and heuristic decision making

Empirical research on human decision making finds that individual behavior depends on the framing and context of the decision (Tversky and Kahneman, 1974). Human decision making is characterized deviations from the normative standards of the rational choice model, so-called *cognitive biases*, challenging the understanding that rational choice theory serves not only
as a normative benchmark, but also as a descriptive model of individual decision making. Biases can be the result of time-limited information processing (Hilbert, 2012), heuristic decision making (Simon, 1956), or emotional influences (e.g., wishful thinking, Babad and Katz, 1991; Loewenstein and Lerner, 2003). *Bounded rationality theory* assumes that human decision making is constrained by *cognitive* and computational capabilities of the agents, additionally to the constraints imposed by the environment and the available information about it (Simon, 1956, 1997). In the economic literature, non-transitive preferences,





time-inconsistent discounting and deviations from expected utility that we already introduced in the previous subsection are often also considered as boundedly rational (Gintis, 2009). Boundedly rational agents can be considered as *satisficers* that try to find a satisfying action in a situation given their available information and cognitive capabilities (Gigerenzer and Selten, 2002).

Constraints on information processing imply that agents do not compute the utility of every possible option in complex decision situations and choose the one with maximal utility. Instead, agent decisions use *heuristics* for judging the available information and choosing actions that lead to the more preferred outcome over less preferred ones. Gigerenzer and Gaissmaier (2011) defines heuristics in decision making as a "strategy that ignores part of the information, with the goal of making decisions more quickly, frugally, and/or accurately than more complex methods." In contrast to so-called 'as if' models of

human decision making that mathematically integrate all available information to mimic the outcome of the decision process, heuristic decision making taps into the process of information gathering and processing and describes it in the form of simple algorithmic rules. Heuristics are considered to be *fast and frugal* in the sense that they do not solve algebraic or optimization problems and evaluate only part of the available information. Consequently, they are well suited for computationally efficient implementations of human decision making in models.

Furthermore, Gigerenzer and Todd (1999) argue that many of the decision theories being used as a benchmark for rationality are not designed for so-called 'large worlds' where information relevant for the decision process is either unknown or has to be estimated from small samples. They question the usefulness of rational choice theory as the normative standard and try to relieve heuristic decision making of its stigma of cognitive laziness, bias and irrationality. In many real world situations, especially when high uncertainties are involved, some decision heuristics perform equally good or even better than more

elaborated decision strategies (Dhami and Ayton, 2001; Dhami and Harries, 2001; Keller et al., 2014). Therefore, it is argued that instead of an all-purpose tool the mind carries an 'adaptive toolbox' of different heuristic decision schemes, that are ecologically rational[6] in particular environments (Gigerenzer and Selten, 2002; Todd and Gigerenzer, 2007).

In general, heuristic rules are formalized either as *decision trees* or *flowcharts* and consist of three building blocks: one for information search, one for stopping information search and one to derive a decision from the information found. They

evaluate a number of pieces of information – so-called cues – to either categorize a certain object or to choose between several options. Many heuristics evaluate these cues in a certain order and make a decision as soon as a cue value allows classification or discriminates between options. This is illustrated by means of the *Take the Best heuristic*: Pieces of information (cues) are compared between alternatives according to a prescribed order until one cue discriminates between the alternatives under consideration. At each step in the cue order of the decision process, some information is searched and evaluated. If it allows

discriminating between the options, the option with the higher cue value is chosen. Else the process moves on to the next cue. This repeats as the process moves down the cue order until a cue is reached where the differentiation between options is possible. For the 'Take the Best' heuristic, the order in which the cues are evaluated is crucial for the result. Other notable examples are *Fast and Frugal Trees* and *satisficing heuristics*. The latter evaluates information sequentially and chooses the

---

[6]Ecological rationality claims that rational decisions should not be made based on rules that are independent of the circumstances (as for example in rational choice theory) but on context-specific ones such as heuristics, making heuristic decision making also a normative choice model.



first option satisfying certain criteria. An overview and explanation of numerous other decision heuristics can be found in the recent review paper by Gigerenzer and Gaissmaier (2011).

Heuristics, especially cue orders, can also be interpreted as encoding norms and preferences in individual decision making as they prioritize features of different options over others and hierarchically structure the evaluation of available information. So far, heuristics have primarily been studied for inferences rather than preferences. Nevertheless, the same frameworks can also be used to describe decisions based on preferences, such as consumer choice (Hauser et al., 2009), voter behavior (Lau and Redlawsk, 2006), or organizational behavior (Loock and Hinnen, 2015; Simon, 1997). Also, recent findings suggest that cue orders can spread via social learning and social influence (Gigerenzer et al., 2008; Hertwig and Herzog, 2009) analogously to norm and opinion spreading in social networks (see Sections 4.3 and 4.4). Therefore, heuristics might be used to shed light on the implications of changing norms and values for individual and collective behavior.

Despite the many upsides of Fast and Frugal decision heuristics, they are not yet commonly applied in dynamic modeling of social-ecological systems. One exception is the description of farmer and pastoralist behavior in a study of origins of conflict in east Africa (Kennedy and Bassett, 2011). However, as the following example shows, decision trees can be used to model decision making in agent-based simulations of land-use change (Deadman et al., 2004). The model describes colonist household decisions in the Amazon rainforest. Each household is a potential farmer who first checks whether a subsistence requirement is met. If this is not the case, the household farms annual plants. If the subsistence requirement is met, the household checks the quality of the soil. In the case of acidic soil, it plants perennials. In the case of non-acidic soil, it plants pasture and breed livestock. If the activities are not affordable, the household does not farm at all. The model shows how simple heuristic decision trees can be used to simplify complex decision processes and represent them in an intelligible way. However, the example also shows the many degrees of freedom in the construction of heuristics, pointing at the difficulty to obtain these structures from empirical research.

Heuristics are a promising tool for including individual human decision making at the micro-level into Earth system models because they can capture basic crucial choices in a computationally efficient way. In order to describe the long-term evolution of preferences and values, which might play an important role for human influences on the Earth system, heuristics could also be used to model meta-decision of preference and value adoption. However, in contrast to fully rational decision making, it can be very challenging to aggregate heuristic decision making to higher organizational levels. Therefore, computational methods like agent-based modeling are needed to explore the aggregate outcomes of such decision processes, which has implications for the possible analyses (see Section 5.5).

### 3.3 Learning theory

The approaches discussed in the previous two subsections mainly took the perspective of a forward-looking agent. Rational or boundedly rational actors optimize future payoffs based on information or beliefs about how their behavior affects future payoffs, while the procedures to optimize may be more or less bounded. However, these techniques do not specify how the information is acquired and how the beliefs are formed. Therefore, another branch of modeling focuses on backward-looking behavior: an agent learned in the past that a certain action gives a reward (or feels good) and therefore the agent repeats its



behavior. Computational learning theory focuses on this narrow understanding of learning. It can help to capture the adaptivity of agent behavior and is particularly suited for modeling behavior under limited information.

*Reinforcement learning* is a modeling approach that captures how an agent maps environmental conditions to desirable actions in a way that optimizes a stream of rewards (and/or punishments). The obtained reward depends on the state of the

environment and the chosen action, but may also be influenced by chosen actions and environmental conditions in the past. According to Macy et al. (2013), reinforcement learning differs from forward-looking behavioral models regarding three key aspects: (i) Because agents explore the likely consequences and learn from outcomes that actually occurred rather than those which are intended to occur but only with a certain probability, reinforcement learning does not need to assume that the consequences are intended. (ii) Decisions are guided by rewards fostering approach or punishment leading to avoidance rather

than utilities. (iii) Rather than optimization, decisions rules are characterized by stepwise melioration.

The learning process is modeled via a learning algorithm that operationalizes different strategies of trial and error, e.g. by a simple value function or temporal difference (Q-learning) algorithms or artificial neural network approaches (Sutton and Barto, 1998). Some learning algorithms have also been inspired by the process of natural selection (*genetic algorithms*). The learning algorithm has to balance a trade-off between the exploration of actions with unknown consequences and the exploitation of

current knowledge. In order to not having to explore all possible actions by brute force, many algorithms use randomness to include deviations from already learned behavior.

The environment in reinforcement learning problems is often modeled as a Markov decision process (Bellman, 1957). In each of the discrete states of the environment the agent can choose from a set of possible actions. The choice then influences the transition probabilities to the next state. Reinforcement learning is an unsupervised learning technique as opposed to supervised

learning, which requires that optimal responses are presented and therefore trained with an external correction. Therefore, it is suitable to model the learning of agents. As an illustration from the land-use context, consider a farmer adapting her planting practices to new climatic conditions by adjusting the timing of sowing, irrigation and harvesting. Without the possibility to acquire knowledge through other channels, she would experiment in some way with the possible adjustments and evaluate how they change the yield (her reward). Eventually, by a trial-and-error process her yield would on average increase.

A standard approach to model the acquisition of *subjective probabilities* associated with the consequences of actions is *Bayesian learning*, which has also been applied to reinforcement learning problems (Vlassis et al., 2012). Starting with some prior probability (e.g. from some high-entropy "uninformative" distribution) $P(h_i)$ that some hypothesis $h_i$ about the relation of actions and outcomes is true, new information or evidence $P(E)$ is used to update the subjective probability with the posterior $P(E|h_i)$ calculated with Bayes' theorem: $P(h_i|E) = P(E|h_i)P(h_i)/P(E)$ (Puga et al., 2015). The most probable

hypothesis can then be chosen to determine further action.

Combining various approaches to model the acquisition of beliefs through learning, the formation of preferences and different decision rules with further insights from psychology and neuroscience has led to the development of very diverse and detailed behavioral theories which are often formalized in *complex cognitive architectures* (Balke and Gilbert, 2014). These approaches can also be used to model human behavior but we will not discuss them in detail here because of their complexity

and diverse formalization.



**Table 2.** Summary table for individual behavior and decision making

| Theories | Key considerations | Strengths | Limitations |
|---|---|---|---|
| Optimal decisions in rational choice: Individuals take the decision that maximizes their expected utility given economic, social and environmental constraints | What are agent's preferences? Which information (and beliefs) do they have? | Highly researched theory with strong theoretical foundation and many applications | Individuals assumed to have strong capabilities for information processing and perfect self-control |
| Bounded rationality and heuristic decision making: Individuals have biases and heuristic decision rules that help them navigate complex environments effectively | Which cue order is used to gather and evaluate information? When do agents stop gathering more information and decide? | Simple decision processes that capture observed biases in decision making | Suitable decision rules highly context dependent |
| Learning: Agents explore possible actions through repeated learning from similar past events | How do agents interact with their environment? What is the trade-off between exploitation of knowledge and exploration of new options? | Captures information and belief acquisition process | High degree of randomness in behavioral changes |

Learning and related theories that emphasize the adaptability of human behavior might be important building blocks to model on the one hand the long-term evolution of human interactions with the Earth system from an individual perspective. On the other hand, they can capture also short-term responses to drastically changing natural environments, which might give insights on behavioral transformations in the future.

5    Table 2 summarizes the approaches that focus on individual human behavior. However, besides the forward- and backward-looking behavior that we introduced in this section, agents may exhibit sideways-looking behavior: agents can copy the behavior of successful others, thereby contributing to a *social learning* process. For this kind of behavior, interactions between different agents are crucial. This will be the focus of the next section.

## 4    Modeling interactions between agents

10    In the previous section, we discussed modeling approaches that focus on the choices of individuals that are confronted with a decision in a specified situation. In contrast, this section focuses on interaction between individuals or groups of agents, decisions where actors respond to each other. We review different kinds of direct interactions and techniques to model them.



Direct interactions are usually local and take place between two (or sometimes a few) agents. Indirect interactions that can happen at broader scales, for instance mediated through price mechanisms on markets, will be discussed as part of Section 5 on aggregation.

The section starts with reviewing strategic interactions as modeled in classical game theory and dynamic interactions within evolutionary approaches and social learning. Then, we address models of social influence that are used to study opinion and preference formation or the transmission of cultural traits, i.e. culturally significant behaviors. Finally, we discuss how interaction structures can be modeled on networks.

## 4.1 Strategic interactions between rational agents: classical game theory

Game theory focuses on situations of "strategic interdependence", decision problems where the utility that a decision-maker (called player) gets does not only depend on her own decision but also on the choices of others. These are often situations of conflict or cooperation. Players choose an action (behavioral option, control) based on a *strategy*, i.e. a rule specifying which action to take in a given situation. *Classical game theory* explores how rational actors identify strategies, usually assuming the rationality of other players. However, rational players can also base their choices on beliefs about others players' decisions, which can lead to an infinite regress of mutual beliefs about each others' decisions.

Formally, a game is described by what game-theorists call a *game form* or *mechanism*. The game form specifies the actions $a_i(t)$ that agents can choose at well-defined time points $t$ from an *action set* $A_i(t)$ that may vary over time, having to respect all kinds of situation-dependent rules. The game form may furthermore allow for communication with the other agent(s) (*signaling*) or binding agreements (*commitment power*). Simple social situations are typically represented as "normal form games" by a *payoff matrix* specifying the individual utilities[7] for all possible action combinations, while more complex situations are modeled as stepwise movement through the nodes of a decision tree or game tree (Gintis, 2009).

In particular, classical game theory assumes that players form consistent beliefs about each others' unobservable behavior. They assume that the other's behavior results itself from an optimal strategy. Because the multi-player inter-temporal optimization often leads to recursive relationships between beliefs and strategies, solving complex classical games becomes quite difficult. Such problems often have several solutions, called *equilibria* (not to be confused with the steady-state meaning of the word) and call for sophisticated nonlinear fixed-point solvers (Harsanyi and Selten, 1988). Only in special cases, e.g. where players have complete information and moves are not simultaneous but alternating, game-theoretic equilibria can easily be predicted by simple solution concepts such as backwards induction (Gintis, 2009). In other cases, one can identify strategies and belief combinations consistent with the following two assumptions. First, each player eventually chooses a strategy that is optimal given her beliefs about all other players' strategies (rational behavior). Second, each player's eventual beliefs about other players' strategies are correct (*rational expectations*). The solutions are called *Nash equilibria*. However, many games have multiple Nash equilibria, and the question of which equilibrium will be selected arises.

Therefore, game theorists try to narrow down the likely strategy combinations by assuming additional forms of consistency and rationality (see Aumann, 2006) such as consistency over time (*sequential and subgame perfect equilibria*), stationarity over

---

[7]Note that despite the term "payoff" matrix, these utilities are unexplained attributes of the agents and need not have a relation to monetary quantities.



time (*Markov equilibria*), and stability against small deviations (*stable equilibria* Foster and Young, 1990; Mäs and Nax, 2016), coordinated deviations by groups (*correlated, coalition-proof, and strong equilibria*), or small random mistakes (*trembling-hand-perfect* and *proper equilibria*, Harsanyi and Selten, 1988). After a plausible strategic equilibrium has been identified, it can then be used in a simulation of the actual behavior resulting from these strategies over time, possibly including noise and

mistakes.

As an example in the land use context, consider two farmers living on the same road. They get their irrigation water from the same stream. A dispute over the use of water emerges. Both may react to the actions of the other in several turns. The upstream farmer located at the end of the road may increase or decrease her water use and/or pay compensation for using too much water to the other. The downstream farmer at the entrance of the road may demand compensation or block the road and thereby cut

the access of the upstream farmer to other supplies. Either of them may appeal to the magistrate or apologize to the other and the magistrate may set quotas or impose fines. This results in a complex game tree that encodes which actions are feasible at which moment and what are the consequences on players' utilities.

The magistrate in the example might not know about the farmers' actions before one of them appeals to him and the other farmer might not know about the appeal until the magistrate acts. If these are the only relevant constraints and it is possible

to explicitly model the information and options of the players at each time point, a classical game theoretical analysis makes sense, determining the rational strategies that the farmers would follow.

Classical game theory is widely applied to interactions in market settings in economics (see also Section 5.2), but increasingly also in the social and political sciences to political and voting behavior in *public and social choice theory* (see e.g., Ordeshook, 1986; Mueller, 2003, and Section 5.1). For example, public choice theory studies strategic interactions between

groups of politicians, bureaucrats and voters with potentially completely different preferences and action sets under constraints by existing law.

While many simple models of strategic interactions between rational and selfish agents will predict only low levels of *cooperation*, more complex models can well explain how bilateral and multilateral cooperation, *consensus*, and stable social structure emerges (Kurths et al., 2015). This has been shown in diverse contexts such as individual bilateral interactions in large

groups (e.g., Helbing and Yu, 2009), bilateral international relations (e.g., Aumann, 2006), multiplayer public goods problems (e.g., Heitzig et al., 2011), group decision making (Heitzig and Simmons, 2012), coalition formation on networks (Auer et al., 2015), and international climate policy (e.g., Heitzig, 2013).

To model relevant decision processes in the Earth system, classical game theoretic analysis could be used for describing strategic interactions between agents which could be assumed highly rational and well informed, i.e. international negotiations

of climate agreements between governments, bargaining between social partners or monopolistic competition between firms. Similarly, international negotiations and their interactions with domestic policy can also be framed as two- or multilevel games (as in some models of political science, e.g., Putnam, 1988; Lisowski, 2002). Furthermore, social choice theory could be used to simulate simple voting procedures that (to a certain extent) determine the goals of regional or national governments.





### 4.2 Interactions with dynamic strategies: evolutionary approaches and learning in game theory

In game theoretic settings, complex individual behavioral rules are typically modeled as *strategies* specifying a behavior for each node in the game tree. Consider as an example the repeated version of the Prisoners' Dilemma in which each of two players can either "cooperate" or "defect" in each period (e.g., Aumann, 2006). A typical complex strategy in this game could

involve reciprocity (defect temporarily after a defection of your opponent), forgiveness (ever so often not reciprocate), and making up (don't defect again after being punished by a defection of your opponent after your own defection).

Many or even most nodes of a game tree will not be visited in the eventual realization of the game and strategies may involve deliberate randomization of actions (e.g. tossing a coin). Therefore, strategies are (like preferences) principally unobservable (only actual behavior is), and assumptions about preferences and strategies are unfalsifiable and hard to validate. For this and

other reasons, often several kinds of additional assumptions are made that constrain the set of strategies further that a player can choose, e.g., assuming only very short memory or low farsightedness ("myopic" behavior) and disallowing randomization, or allowing only strategies of a specific formal structure such as heuristics (see Section 3.2).

The above-mentioned water conflict example (see Section 4.1) bears some similarity to the repeated prisoners' dilemma in that the farmers' possible actions can be interpreted as either defective (using too much water, blocking the road, appealing to

the magistrate) or cooperative (not do any of this, compensate for past defections). Assuming different levels of farsightedness may thus lead to radically different predictions since myopic players would much more likely get trapped in a cycle of alternating defections than farsighted players. The latter would recognize some degree of forgiveness as that maximizes long-term payoff and would thus desist from defection with some probability. In any case, both farmers' choices may be modeled as depending on what they believe the other will likely do or how she will react.

Evolutionary approaches in game theory study the interaction of different strategies and analyze which strategies prevail on a population level as a result of selection mechanisms. Thus, in contrast to classical game theory, evolutionary approaches focus on the dynamics of strategy selection in populations. The agent's strategies may be either hardwired, acquired or adapted by learning (e.g., Fudenberg and Levine, 1998; Macy and Flache, 2002). Although many evolutionary techniques in game theory are used in biology to study biological evolution (variation through mutation, selection by fitness and reproduction with inheritance), *evolutionary game theory* can be used to study all kinds of strategy changes in game theoretic settings, for

instance cultural evolution (transmission of memes), social learning through imitation of successful strategies or the emergence of cooperation (Axelrod, 1984, 1997).

In an evolutionary game, a population of agents is divided into factions with different strategies. They interact in a formal game (given e.g. by a payoff matrix, see Section 4.1), in which their strategy results in a fitness (or payoff/utility). The factions

change according to some replicator rules that depend on the acquired fitness. This can be modeled using different techniques. Simple evolutionary games on well-mixed large populations can be described with replicator equations. The dynamics describing the relative change in the factions of strategy $i$ is proportional to the deviation of the fitness of this faction from the average fitness (Nowak, 2006).



Alternatively, the behavior resulting from evolutionary interactions is often easy to simulate numerically as a discrete-time dynamical system even for large numbers of players if the individual action sets are finite or low-dimensional and only certain simple types of strategies are considered. This type of agent-based model (see Section 5.5) simply implements features such as mutation or experimentation and replication via strategy transfer (imitation) at the micro-level. Combined with (adaptive)

social network approaches (see Section 4.4), the influence of interaction structure can also be studied (Szabó and Fáth, 2007; Perc and Szolnoki, 2010). The steady states of evolutionary games are usually characterized by so-called *evolutionary stable strategies* or *stochastically stable equilibria*. A population which adopts an evolutionary stable strategy cannot be invaded by other, initially rare strategies. If the steady state is stable for finite populations or noisy dynamics, stable equilibria are called stochastic.

In our water conflict example, the farmers could use a heuristic strategy (see Section 3.2) that determines how much water they extract given the actions of the other. The evolution of the strategies could either be modeled with a learning algorithm, repeating the game again and again. Alternatively, to determine feasible strategies in an evolutionary setting, a meta-model could consider an ensemble of similar villages consisting of two farmers and a magistrate. The strategies of the farmers would then be the result of either an imitation process between the villages, or of an evolutionary process, assuming that less successful

villages die out over time.

Evolutionary approaches to game theory are a promising framework to better understand the prevalence of certain human behaviors regarding interaction with the Earth system. This is especially interesting regarding the modeling of long-term cultural evolution and changes in individual's goals, beliefs and decision strategies or the transmission of endogenous preferences (Bowles, 1998).

## 4.3  Modeling social influence

Another strong force in human interaction is social influence, a process in which individuals adjust their opinions, beliefs, preferences, or behavior after interacting with others. Humans exert influence on each other for various reasons. They may be convinced by persuasive arguments (Myers, 1982), aim to be similar to esteemed others (Akers et al., 1979), are unsure about what is the best behavior in a given situation (Bikhchandani et al., 1992), or perceive social pressure to conform with others

(Wood, 2000; Festinger et al., 1950; Homans, 1951).

*Models of social influence* allow studying the outcomes of repeated influence in social networks and have been used to explain the formation of consensus, the development of mono-culture, the emergence of clustered opinion distributions, and the emergence of opinion polarization, for instance. Models of social influence are very general and can be applied to any setting where individuals exert some form of influence on each other. However, seemingly innocent differences in the formal

implementation of influence in models can have decisive effects on the model outcomes. In the following, we list important modeling decisions that have been shown to have significant implications.

A first question is *how* agent attributes are influenced by interactions with others. Classical models incorporate influence as averaging, which means that interacting individuals become more similar over time (Friedkin and Johnsen, 2011). *Averaging* is an accepted and empirically supported model of influence resulting, for instance, from social pressure that an actor exerts





on someone else (Takács et al., 2016). In other contexts, averaging is debated (Myers, 1982; Mäs and Flache, 2013; Mäs et al., 2013; Myers and Lamm, 1976; Vinokur and Burnstein, 1978). For instance, some models of opinion influence assume that opinion influence results from argument communication (Mäs and Flache, 2013; Mäs et al., 2013). When actors with similar opinions interact in these models, their opinions do not always converge. Instead, they turn more extreme as the interaction

partner provides them with new arguments that support the own opinion. Likewise, some models assume different forms of averaging: Rather than following the arithmetic average of all opinions, actors might only consider the majority view (mode) in their network (Nowak et al., 1990). For example, a farmer considering on which date to best till his field might either take the date which most of his neighbors think is best or simply take the average of all the proposed dates.

      Second, one could ask whether there are just one or several *dimensions of influence*. For instance, it is often argued that

political opinions are multi-dimensional and cannot be captured by the one-dimensional left-right spectrum. Explaining dynamics of opinion polarization and clustering turned out to be often more difficult when multiple dimensions are taken into account (Axelrod, 1997). Additionally, model predictions often depend on whether the influence dimension is a *discrete* (see e.g., Axelrod, 1997; Mark, 1998; Carley, 1991; Galam, 2005; Nowak et al., 1990) or a *continuous variable* (see e.g., DeGroot, 1974; French, 1956; Lehrer, 1975; Friedkin and Johnsen, 2011). Models of individuals' decisions about certain policies often

model the decisions as binary choices (Sznajd-Weron and Sznajd, 2000; Martins, 2008). However, binary scales fail to capture that many opinions vary on a continuous scale and that differences between individuals can therefore increase also on a single dimension (Barker and Lawrence, 2006; Dalton et al., 1998; Feldman, 2011; Jones, 2002; Mäs and Flache, 2013; Stroud, 2010). Therefore, models that describe opinion polarization usually treat opinions as continuous attributes. The opinion on a land reform can, for instance, be modeled as a binary variable (approval or rejection) whereas the willingness to support it

could be better described by a continuous variable from strong support to strong opposition.

      A next critical question is whether agents' characteristics can travel in different directions from one person to another, i.e. if the influence is directional. In models of opinion dynamics, for example, influence is often *bi-directional* in the sense that an actor who exerts influence on someone else is also influenced by the other (Macy et al., 2013; Mäs et al., 2010). But the influence can also be only possible in one direction or the strength of influence can be asymmetric. Furthermore, the influence

may be *multilateral* or *dyadic*, i.e. only between two interaction partners. Model outcomes often depend on whether influence is modeled as an event involving a dyad of actors or multiple contacts for opinion update (Parisi and Cecconi, 2003; Flache and Macy, 2011; Lorenz, 2005; Huckfeldt et al., 2004). In models that assume binary influence dimensions, for instance, dyadic influence implies that an agent copies a trait from her interaction partner. When influence is multilateral, agents aggregate the influence exerted by multiple interaction partners (using e.g. the mode of the neighbors' opinions), which can imply that agents

with rare traits are not considered even though they would have an influence in the case of dyadic influence events. It has been demonstrated that this can have important consequences on equilibrium predictions (Flache and Macy, 2011; Huckfeldt et al., 2004). For example, a farmer seeking advice whether to adopt a new technology can either consult his friends one after another or all together, likely leading to different outcomes if they have different opinions on the matter.

      Social influence is a strong force but it is not plausible to assume that agents never deviate from the influence of their

contacts. The exact model of these *deviations* affects model outcomes and can introduce a source of diversity into the model



(Mäs et al., 2010; Pineda et al., 2009; Kurahashi-Nakamura et al., 2016). For instance, some models of continuous opinion dynamics include deviations as Gaussian noise, i.e. random values drawn from a normal distribution. In such a model, noise implies that opinions in homogeneous subgroups will randomly fluctuate, which aggregates to collective random walks of subgroup members through the opinion space. When two subgroups happen to adopt similar opinions, influence will lead to a fusion of subgroups that would have remained split in a model without deviations (Mäs et al., 2010). In other contexts, deviations are better modeled by uniformly distributed noise, assuming that big deviations are as likely as small ones. This can help to explain for instance the emergence and stability of subgroups with different opinions, that do not emerge in settings with Gaussian noise[8] (Pineda et al., 2009). In the context of land use, the opinion dynamics regarding a land reform may not only be determined by the interactions between individual agents but may also be influenced by mass media that randomly shifts individual's opinions.

To model transitions regarding norms and lifestyle changes to sustainable consumption, admissible resource use and emissions, as well as technology adoption at a micro-level, models of social influence are an important tool. These mechanisms can furthermore be combined with changes in social structure and be modeled via adaptive networks, as we show in the next section.

## 4.4 Modeling the evolution of interaction structure: adaptive network models

In most of the models discussed in the previous section, the social network can be formally modeled as a graph (the mathematical notion for a network): a collection of nodes that are connected by a collection of links. In this mathematical framework, nodes (vertices) represent agents and links (edges) between the agents indicate that agents interact by communicating and exchanging information or form social relationships. Agents can only interact and thus influence each other if they are connected by a link in the underlying network. Note that network models of agents can be understood as a special case of agent-based models, which we discuss in more detail in Section 5.5.

Classical social-influence models study the dynamics of influence on static networks, assuming that agents are always influenced by the same subset of interaction partners (Abelson, 1964; DeGroot, 1974; French, 1956; Harary, 1959; Friedkin and Johnsen, 2011). As discussed above, these networks can be directed or undirected, but their structure does not change over time. In social influence models on static networks, fully connected populations will usually reach perfect consensus in the long run. However, it depends on the duration of the modeled processes whether the assumption of flexible network ties is plausible. For instance, in an organization where individuals have fixed position in organizational subunits networks appear to be less flexible than in on-line social networks.

Especially when modeling social processes over longer time scales, it is reasonable to assume that the social network is dynamic, i.e. that its structure evolves over time. This time evolution can be independent of the dynamics on the network encoded in temporal networks (Holme and Saramäki, 2012). However, for many social processes, it can be assumed that the structure of the social network and the dynamics on the network (e.g. social influence) interact. *Adaptive network models* make

---

[8]Gaussian noise needs to be very strong to generate enough diversity for the emergence of subgroups with different opinions. However, when noise is strong, subgroups will not be stable.



the removal of existing and the formation of new links between agents dependent on attributes of the agents. Thus they build on the insight that the social structure influences the behavior, opinion or value systems of individual actors which in turn drives changes in social structure (Gross and Blasius, 2008).

Local update rules for the social network structure and the agent behavior can be chosen very flexibly. The rules can be
deterministic or stochastic and described for example by discrete maps, ordinary differential equations or logical operations (related to cellular automata). Changes in agent behaviors may be governed by rules such as random or boundedly rational imitation of the behavior of network neighbors (see above). Relevant update rules for network structure describe processes such as homophily, where agents with similar states tend to form new links between each other while breaking links with agents having diverging states (Wimmer and Lewis, 2010; McPherson et al., 2001; Lazarsfeld and Merton, 1954). This common assumption
is based on the insight that agents tend to be influenced by similar others and ignore those sources who hold too distant views (Axelrod, 1997; Carley, 1991; Hegselmann and Krause, 2002; Deffuant et al., 2005). In adaptive network models, homophily in combination with social influence generates a positive feed-back loop: influence increases similarity, which leads to more influence and so on. Such models can explain for instance the emergence and stability of multiple internally homogeneous but mutually different subgroups. Other applications allow to understand the presence of social tipping points in opinion forma-
tion (Holme and Newman, 2006), epidemic spreading (Gross et al., 2006) in systems of networked agents, the emergence of cooperation in social dilemmas on co-evolutionary networks (Perc and Szolnoki, 2010) and the co-evolution of multilateral cooperation (coalition formation) with social networks (Auer et al., 2015). The dynamical interaction in adaptive network models can give rise to complex and nonlinear co-evolutionary dynamics such as phase transitions (Holme and Newman, 2006; Auer et al., 2015), multi-stability (Wiedermann et al., 2015), oscillations in both agent states and network structure (Gross et al.,
2006), and subtle but robust changes in social structure (Schleussner et al., 2016).

While adaptive networks have so far mostly been applied to networks of agents representing individuals, the framework can in principle be used to model co-evolutionary dynamics on various levels of social interaction as introduced in Table 1. For instance, global complex network structures such as financial risk networks between banks, trade networks between countries, transportation networks between cities and other communication, organizational and infrastructure networks can be modeled
(Currarini et al., 2016). Furthermore, approaches as multi-layer and hierarchical networks or networks of networks allow modeling the interactions between different levels of a system (Boccaletti et al., 2014).

As an illustration for an application in the land-use context, consider a community of farmers described by a network of social relations. The farmers are faced with the choice to adopt a new agricultural technology which is potentially more productive, but this is uncertain. If the social acquaintances of a farmer successfully test the new technology, she is more likely
to adopt it herself. However, if the adoption is not successful, she might form relationships with other farmers that have not yet adopted the new technology. In this way, rich model dynamics can emerge, that may either lead to a full adoption of the new technology or a segregation of the community into a group with and another without the new technology, depending on the model parameters.

In the context of long time scales in the Earth system, the time evolution of social structures that determine interactions
with the environment are particularly important. Adaptive networks offer an interesting approach to modeling transformative



**Table 3.** Summary table for agent interactions.

| Approaches and frameworks | Key considerations | Strengths | Limitations |
| --- | --- | --- | --- |
| Classical game theory: strategic interactions between rational agents | What is the game structure (options, possible outcomes, timing, information flow) and what are the players' preferences? | Elegant solutions for low-complexity problems | Agents cannot change the rules of the game |
| Evolutionary game theory: competition and selection between hardwired strategies | Which competition and selection mechanisms are there? | Can explain how dominant strategies come about | Agent strategies are modeled as hard-wired (no conscious strategy change) |
| Social influence: agents change their beliefs, preferences and opinions | What influence mechanisms are dominant? Is social influence multilateral or dyadic? How large are deviations? | Allows to model social learning, preference formation, and hearding behavior | Local dynamics often stylized |
| Network theory: changing social interaction structure | Is the social network static or adaptive? How do agents find new neighbors? How much randomness and hierarchy is in the structure? | Mathematical formalization to model co-evolution of social structure with agent attributes | Micro-interactions mostly dyadic and schematic |

change with deep structural imprints on internal connectivity of a complex system (Lade et al., 2016) such as an alleged great transformation to sustainability (Schellnhuber et al., 2011) that may involve the transition from centralized to decentralized infrastructure network structures.

Table 3 summarizes the different modeling approaches that focus on agent interactions in human decision making and
5   behavior. These interactions occur between two or several agents. For including the effect of these interactions into Earth system models, their aggregate effects need to be taken into account as well. Therefore, we introduce in the next section approaches that allow to aggregate individual behavior and local interactions and to study the resulting macro-level dynamics.

## 5   Aggregating behavior and decision making and modeling dynamics at the system level

So far, we focused on theories and modeling techniques that describe decision processes and behavior of single actors, their
10   interactions and the interaction structure. This section builds on the previously discussed approaches and highlights different



methods how the behavior of an ensemble or group of agents might be aggregated. This is an important step if models shall describe collective decision making and behavior in the context of Earth system modeling. In general, aggregation can take place on all levels introduced in Section 2 and summarized in Table 1. Aggregation techniques link modeling assumptions at one level (often called the micro-level) to a higher level (the macro-level). They therefore enable the analysis of emergent
macro-level outcomes and help to transfer models from one scale to another.

In this section, we describe different approaches that are used to make this connection: On the one hand, analytical tools allow representing groups of individual agents through some macro-level or average characteristic, often using simplifying assumptions regarding the range of individual agents' characteristics. On the other hand, simulations describe individual behavior and interactions and computational methods allow to study the resulting aggregate macroscopic dynamics.

The question how to aggregate micro-processes to macro-phenomena is not specific to modeling human decision making and behavior. Aggregation of individual behavior and the resulting description of collective action, such as collective motion, is also an ongoing challenge in the natural sciences (see e.g., Couzin, 2009). Specific assumptions about the individual behavior and agent interactions have consequences for the degree of complexity of the macro-level description. For instance, if agent goals and means do not interact, the properties of single agents can often be added up. If, on the contrary, agents influence each
other's goals or interact via the environment, complex aggregate dynamics can arise.

The following sections discuss the specificities for aggregating human decision making and behavior and notable applications in models in the global environmental change context. As some aggregate dynamics are difficult to reduce to micro-behavior and interactions, the section concludes with discussing important macro-level approaches with applications in Earth system modeling.

## 20  5.1  Aggregation of preferences: social welfare and voting

The original micro-level framework of rational choice is often applied to model the behavior of groups of individuals at all levels introduced in Table 1. *Social choice theory* explores how individual preferences can be aggregated to social welfare, a measure of collective desirability of an outcome. Furthermore, it analyzes how group choices can be determined best in voting procedures, in which group members choose between different options and the collective choice is determined by some formal
rules.

As in Section 3.1, utility functions can form the basis for modeling *preferences of groups* by a "social utility". Individual utilities can be aggregated into a *social welfare function* by making the assumption that individual agents have a common scale-measurable unit of utility ("util"), which represents an amount of satisfaction, happiness or sometimes simply a monetary value. Most often, modelers use the linear and inequality-neutral utilitarian welfare function, taking the average over the $N$
individuals in the group, $U(x) = \sum_i u_i(x)/N$. Sometimes this is motivated by the assumption that groups may redistribute utility internally to mitigate inequality (*transferable utility*). However, it is highly debated that utilities of different individuals can be compared and substituted in the sense that a drop in collective welfare resulting from an actor's decrease in utility can be compensated by increasing the utility of another actor. Though, when only considering ordinal preference relations instead



of cardinal (scale-measurable) utility into account, general statements about aggregated preferences are very limited (Arrow, 1950).

Inequality-averse groups can be modeled using $U(x) = \sum_i f(u_i(x))/N$ for some concave function $f$, or via welfare functions based on inequality measures such as the Gini-Sen welfare function, $U(x) = \sum_{i,j} min(u_i(x), u_j(x))/N^2$ (Dagum, 1990), the Atkinson-Theil-Foster welfare function (Dagum, 1990) or, in the extreme case, the egalitarian welfare function $U(x) = \min_i u_i(x)$. In economic contexts, welfare functions are often based on monetary values such as wealth, income or total value of consumption. Defining suitable group preferences becomes especially complicated when the group composition or size $N$ changes over time as in intergenerational models (e.g., Millner, 2013).

Once a social welfare function is constructed, one may identify the social welfare associated with different collective actions and choose the one with the maximal value. The social welfare is used for instance as a criterion to evaluate which policy in a bundle of options leads to the social optimum. Welfare maximization reduces to *cost-benefit analysis* if the utilities are simply added up and are equated with monetary values (Feldman and Serrano, 2006). An alternative to such policy evaluation tools is multi-criteria decision making (Huang et al., 2011). However, cost-benefit analysis remains one of the most applied decision models to normatively evaluate policies and can therefore also be used to model government decisions descriptively.

An example from the land-use context illustrates the concept of "social utility". In a village farmers grow crops that each need specific amounts of water. Water management policies thus affect the incomes of the farmers in different ways. The effect of a policy on the village can be evaluated using either the average or the minimal income of the farmers or some more complex aggregation. Then, the policy should be taken that maximizes this indicator of social welfare. Analogous criteria might be used in policy-making on higher levels of social organization from towns to countries. In complex organizations, however, the actual decision might be non-optimal and a more explicit modeling approach of actual decision procedures might describe the decisions better, e.g. using a game-theoretical model with voting or bargaining procedures.

In *voting theory*, a set of *voters* partitioned into factions with similar preferences can decide over the group's joint actions by means of a formal bargaining or voting protocol. The protocol is designed to find a compromise between the factions' preferences (*cooperative game theory*). Under different voting methods, subgroups may dominate the decision or the group may be able to reach a compromise (cp. Heitzig and Simmons, 2012), also depending on the individual's strategies. Voting methods can be seen as an aggregation mechanism for individual (and possibly heterogeneous) preferences. In the above example, the farmers may not agree on a social welfare measure that a policy should optimize but instead on a formal protocol that would allow them to determine a policy for water usage that is acceptable for all.

## 5.2 Aggregation via markets: economic models and representative agents

Nowadays, a major part of the relevant interaction of societies with the Earth system is closely linked to production and consumption organized on markets. Markets do not only mediate between the spheres of production and consumption, they can also be seen as a mechanism to aggregate agents' decisions and behavior. Economic theory builds on rational choice theory to ask how goods and services are allocated and distributed among the various activities (sectors of the economy) and agents (firms, households, governments) in an economy. Goods and services may be consumed or can be the input factors to



economic production. Input factors for production are usually labor and physical capital, but can also include financial capital, land, energy, natural resources and intermediate goods. In markets, the coordination between *demand* and *supply* of goods is mediated through *prices* that are assumed to reflect information about the abundance or scarcity of goods. Economics compares different kinds of market setting (e.g., auctions, stock exchanges, international trade) with respect to efficiency criteria.

Microeconomic models analyze the conditions for an equilibrium between supply and demand on a single market and between all markets in an economy. A central idea is that an economy is characterized by *decreasing marginal utility and production*: The additional individual utility derived from the consumption of one additional unit of some good (or of one additional hour of leisure) is declining. Similarly, it is assumed that the additional amount of a production derived from one additional unit of some input factor is declining with its absolute amount. The output of the production process is described as

a *production function* that has input factors as arguments. Decreasing marginal utility and production implies that utility and production functions are concave in their arguments (i.e., they have a negative second derivative). Assuming that there is *perfect competition*, resources and goods would be allocated in a *Pareto-efficient* way so that no further redistribution is possible that benefits somebody without making somebody else worse off (Varian, 2010). However, in markets dominated by only a few or very heterogeneous agents perfect competition cannot be assumed, and price wars, hoarding, and cartel formation can occur.

Such situations can be described in models of oligopoly, bargaining or monopolistic competition but are sometimes difficult to integrate into macroeconomic frameworks.

The idea of the *market equilibrium* can most easily be understood by the associated prices: The rational market participants trade goods as long as there is somebody who is willing to offer a unit of some good at a lower price than what somebody else is willing to pay for it (bid price). In a competitive setting, offer prices will go up and bid prices down after each trade because

of decreasing marginal utility that the agents can derive from obtaining more of the same products. Under some conditions, one can show that this leads to the emergence of an equilibrium price for each good to which all local offer and bid prices converge as the market is cleared and supply meets demand.

Macroeconomic models are often built on this micro-economic theory incorporating decision making of firms and households with the representative agent approach. A representative agent stands for an ensemble of identical agents or an average

agent of a population that can be heterogeneous to some degree. However, an underlying assumption is that heterogeneities and local interactions cancel out for large numbers of agents. The behavior of these representative households and firms is usually modeled as utility and profit maximization, respectively. Furthermore, they are assumed to act as if there would be perfect competition and they had no *market power*, i.e. that they take the prices of inputs and outputs as given and cannot influence them. The dynamics of the economy is then the result of the optimizing behavior under various constraints.

In simple *general equilibrium* (GE) models, different sectors of the economy are modeled by representative firms and the demand is determined by one or several representative household. The representative agents interact on perfect markets for all factors of production and consumer goods. In other words, it is assumed that all sectors pay the same wage for an hour of some type of labor, the same interest on financial capital, and the same price for any other input factor. These prices equal the value of what they are able to produce additionally by using one more hour of labor or one more unit of the respective input

factor, i.e. their *marginal product*. The household can consume goods worth the capital and labor income it receives. This leads





to a system of nonlinear algebraic equations in prices and quantities that may be solved by convex optimization, resulting an allocation of input factors, their prices and the resulting output from it. The implication of these equilibrium arguments are full employment of labor and capital in models that allow for substitution of production factors.

As an example from land use, consider the effect of the introduction of a new technology which requires more capital and less labor for the same production of the same good: In a general equilibrium framework, this could be modeled by a closed agrarian sector with two representative agents and a fixed amount of labor and capital. When the representative agent using the old technology has to compete against the one with the new technology, the wages would fall and interest rates would rise until an equilibrium is reached. In this equilibrium, the prices and allocation of input factors are such that the producer with the old technology can produce at the same price as the producer with the new one. An interpretation of this model at the micro-level

is that farmers would switch to the new technology until the prices of input factors adjust to the new demand.

In reality, prices can undergo rapid fluctuations, which challenges the validity of equilibrium assumptions at least in the short run. Models capture deviations from equilibrium by *market imperfections* such as transaction costs, asymmetries in available information and stochastic shocks due to new information as well as changes of production functions due to technological change. Dynamic stochastic general equilibrium (DSGE) models account for imperfections by applying stochastic shocks to

technological developments and prices. They model the expectations of economic agents and the corresponding consumption and investment decisions under uncertainty. Most modern DSGE models also incorporate short-term market frictions such as barriers to nominal price adjustments ("sticky" prices, which have consequences for inflation) and imperfect competition (Wickens, 2008). However, these models still build on the key concept of general equilibrium because they assume that the state of the economy is always near such an equilibrium and market clearance is fast.

In the land-use context, a DSGE approach was used to model land-price dynamics (Liu et al., 2013). The investment decision of the representative firm in the model is not unconstrained as in a general equilibrium framework but constrained by having to provide security in the form of physical capital and land when borrowing financial capital for investment. When subject to several types of shocks (in housing demand, labor supply, or credit availability), the dynamics describes off-equilibrium land price evolution.

In addition to the equilibration between supply and demand through prices, the dynamical evolution of economic quantities, as captured in *Economic growth models*, is important for modeling aggregate human impacts in the Earth system. For example, economic growth is an important driver of energy and resource consumption as well as changes in the agricultural sector (Mundlak, 2000). In standard growth models, a quantity $Y(t)$ of a homogeneous product is produced per time unit according to an aggregate production function depending on productive physical capital $K(t)$, labor and possibly some other inputs[9]. A

part of the output is saved and invested into new capital while the rest can be consumed. The evolution of the capital stock $K$ is described by a differential equation, e.g., $dK/dt = sY(t) - \delta K(t)$. Here, the fraction $s$ of the output is as new capital and the capital depreciates with a rate $\delta$.

---

[9]Standard aggregate production functions are characterized by decreasing marginal productivity and constant returns to scale (i.e., if all the inputs are doubled, the output also doubles).





In the typical *neoclassical growth model* (Ramsey, 1928; Cass, 1965; Koopmans, 1965), the fraction of saving $s(t)$ is endogenously determined by inter-temporal optimization of a representative household. It is assumed that the household maximizes an exponentially discounted utility stream $U(t)$ (compare Section 3.1), which is a function of its consumption $C(t) = (1 - s(t)) Y(t)$ (Acemoglu, 2009). The central decision of the representative household is thus how much of the pro-

duced output it saves and invests at each point in time and therefore cannot consume and enjoy directly. The optimization problem can be solved either computationally by discretization in time or analytically by applying variational calculus (optimal control theory[10]). Besides population growth, the only long-term drivers of growth in the standard neoclassical growth model are changes in the production function representing technological changes, i.e. increases in factor productivity. While technological change is exogenous in standard growth models, it is modeled explicitly in so-called endogenous growth models

(Romer, 1986; Aghion and Howitt, 1998).

The assumption of representative agents in macroeconomic models has theoretical implications that stem from the implicit assumption that the representative agent has the same properties as an individual of the underlying group (Kirman, 1992; Rizvi, 1994): First, the approach neglects that single agents in the represented group have to coordinate themselves, leaving out problems that arise due to incomplete and asymmetric information. Second, a group of individual maximizers does not

necessarily imply collective maximization, challenging the equivalence of the equilibrium outcome. Finally, the representative agent approach may neglect emergent phenomena from heterogeneous micro-interactions (Kirman, 2011).

In spite of the deficiencies of the representative agent approach, its application to markets allows to aggregate behavior in simple and analytically tractable forms. Modelers who wish to describe economic dynamics at an aggregate level can rely on a well developed theory that describe economic growth in a plausible way. In the following section, we will discuss how this

approach is used in combination with the impacts of this economic engine on Earth system.

### 5.3    Modeling of decisions in integrated assessment models: social planner and economic policy

Starting from economic growth and equilibrium models discussed above, environmental economics developed models to account for various environmental externalities in the economy (see e.g., Perman et al., 2003). Externalities are defined as benefits from and damages to the natural system that are not reflected in prices. Such models allow evaluating the extraction

of exhaustible resources, environmental pollution and overexploitation of ecosystems economically. They also help to design economic policies to tackle the associated environmental problems.

*Integrated assessment models* (IAMs) draw on these ideas and combine macroeconomic models with detailed representations of sectors that are closely linked to the environment. They are the most common models that combine both a micro- and a macro-description of human activities at a scales with relevant consequences for the global environment. IAMs couple eco-

nomic activities to environmental variables by incorporating material flows explicitly. For example, $CO_2$ emitted from burning fossil fuels is linked to economic production by carbon intensities and energy efficiency in different production technologies. IAMs usually take the perspective of a social planner, who makes decisions on behalf of society by optimizing a social wel-

---

[10]Optimal control theory deals with the problem of finding the optimal control (often called policy) given by a set of differential equations for the control variables that optimize a certain objective function of a (dynamical) system (under constraints), see for example Kamien and Schwartz (2012).



fare function (see Section 5.1). It is assumed that the social optimum equals the perfect market outcome with a policy that internalizes all external effects.[11]

IAMs are often computational (general or partial) equilibrium models, using large data sets for parametrization and calibration of initial conditions. Regarding modeling technique, IAMs can be broadly categorized with respect to (1) their scope of

representation and (2) their inter-temporal modeling. (1) General equilibrium models represent the whole economy and assume simultaneous market-clearing between all sectors. They often combine a top-down macroeconomic model with bottom-up sectoral models. Partial equilibrium models, on the other hand, only incorporate parts of the economy explicitly, such as the land and energy system. Projections of macroeconomic variables (interest rates, wages, etc.) then drive these sectoral models exogenously. (2) Inter-temporal optimization models allocate investments and consumption optimally over time. They use discounted

social welfare functions as discussed above as objective functions in an optimization procedure. Recursive dynamics models solve an equilibrium for every time step. The dynamics is usually prescribed by difference equations that are derived from considerations about optimal allocation. In these models the inter-temporal allocation is generally non-optimal (Krey et al., 2014; Babiker et al., 2009).

Furthermore, IAMs differ with respect to the representation of technological change, model flexibility in capital realloca-

tion between sectors and regions, and the implementation of trade between subregions. Investment choices between different technology options have long-term effects because the relative prices of technologies can be reduced by induced technological change. Models typically use *constant elasticity of substitution* (CES) production functions that allow for shifts between different (intermediate) products. IAMs represent technological change in different ways: technological parameters such as energy efficiencies or production costs are given as model input or are represented for example by learning-by-doing effects on costs

with increasing installation or endogenous investments in R&D. The implementation of trade between subregions depends on the model type: If the model is solved by global optimization, trade between regions emerges endogenously. Other models determine the trade between regions exogenously.

With respect to their objectives, it is common to distinguish between two categories of IAMs (Weyant et al., 1996): First, policy optimization models (POM) make a complete cost-benefit analysis between the costs of mitigation policies in terms of

consumption or welfare losses and the costs of climate change impacts and adaptation. Thereby, they determine the optimal emission target. The costs are usually represented in highly aggregated damage functions and have led to extensive discussion about the validity of such models. Second, policy evaluation models (PEMs) assess policy options and socio-economic paths with respect to their cost-effectiveness to achieve certain emission targets. They usually have emission targets as constraints in the optimization procedure (see for instance the most recent IPCC report, Clarke et al., 2014).

For the analysis of global land-use, IAMs combine geographical and economic modeling frameworks (Darwin et al., 1996; Lotze-Campen et al., 2008; Hertel et al., 2009; Havlík et al., 2011). These models are used for example to investigate interactions between land allocation and price mechanisms, competition between different land uses (forestry, bioenergy and food production) and trade-offs between agricultural expansion and intensification. They often assume that land-uses can be instan-

---

[11]This argument is based on the second fundamental theorem of welfare economics, see for example Feldman and Serrano (2006, pp. 63–70).



taneously and globally allocated, only constrained by environmental factors such as soil quality and water availability, as well as climate and protection policies.

IAMs differ not only in their technique (mostly optimization) but also in their purpose from Earth system models: They help policy advisors to assess normative paths that the economy could take to reach environmental policy goals. Because

IAMs represent the supply side in much detail, they are used to evaluate investment decisions under different policy choices, for instance between different energy production technologies. Thus, while the policy decision is exogenous to the model, the investments between sectors are only modeled as a reaction to the political constraints. It would be a promising exercise to couple the policy decision in an IAM with some opinion dynamics that depends on the development of the economy. Furthermore, there is already a continuing effort to couple approaches from IAM and ESM (van Vuuren et al., 2012). However,

this could prove very difficult due to the incompatible modeling approaches, especially regarding the treatment of time.

### 5.4    Modeling agent heterogeneity via distributions and moments

As discussed in Section 5.2, the representative agent approach can hardly capture heterogeneity in human behavior and interaction. In this section we describe analytical techniques that allow to capture at least some forms of this heterogeneity. An ensemble of similar agents can be modeled via statistical distributions if the agents are heterogeneous regarding only

some quantitative properties. Such properties could for example be endowments such as income or wealth or parameters in utility functions. In simple models, techniques from *statistical physics* and theoretical ecology can be used to derive a macro-description from micro-decision processes and interactions. For instance, the distribution of agent properties representing an ensemble of agents can be described via a small number of statistics such as mean, variance and other moments or cumulants. The dynamics in form of difference or differential equations of such statistical parameters can be derived by different kinds

of approximations. A common technique is *moment closure* that expresses the dynamics of lower moments in terms of higher order moments. At some order, the approximation is made by neglecting all higher order moments or approximating them by functions of lower-order ones (see e.g., Goodman, 1953; Keeling, 2000; Gillespie, 2009).

To aggregate simple interactions between single nodes in network models, similar techniques can be used to describe the frequencies of particular simple subgraphs with differential equations. In network theory, these approaches are often also called

moment closure, although the closure here refers to neglecting more complicated subgraphs (see e.g., Do and Gross, 2009; Rogers et al., 2012; Demirel et al., 2014). For example, the simple *pair approximation* only considers different subgraphs consisting of two vertices (agents) and one link. To abstract from the finite-size effects of fluctuations at the micro-level in stochastic modeling approaches and arrive at deterministic equations, analytical calculations often take the limit of the agent number going to infinity (in statistical physics called the thermodynamic limit, see e.g., Reif, 1965; Castellano et al., 2009).

The following illustration shows how these techniques could be applied in the land-use context: Consider a model that describes the interaction between farmers who can decide on the amount of fire clearing on their land depending its soil quality and policy choices by a government. The farmers interact on a social network and imitate actions that are profitable under an imposed policy. Such a system can be described by a continuous variable measuring the fraction of farmers that apply fire clearing. The dynamics of this variable would depend on of the success of the different types of farmers, which in turn would





depend on the soil quality of the farmers' land. The latter could be described by the mean and variance of the soil qualities of the two factions of farmers. The resulting system would describe the dynamics of the statistical measures and could be analyzed analytically with methods from statistical physics.

Techniques based on moment closure and network approximations can be used in order to aggregate the dynamics of processes like opinion formation on networks. This could allow to investigate the interplay of such social processes with natural dynamics of the Earth system, e.g. coupled through resource extraction or emissions (cp. Wiedermann et al., 2015).

### 5.5 Aggregation in agent-based models

Agent-based modeling (ABM) is a computational approach to modeling micro-level interactions between individual, autonomous agents and their social and/or biophysical environment and studying their emergent macro-level outcomes (Epstein, 1999; Gilbert, 2008; Heckbert et al., 2010; Edmonds and Meyer, 2013; Hamill and Gilbert, 2016). In agent-based models (ABMs), human behavior is not aggregated to the system level a priori nor is it assumed that individual behavioral diversity can be represented by a single representative agent as in many macroeconomic models (cp. Section 5.2). Instead, population level dynammics emerge from the interactions of heterogeneous agents. ABM is widely used to study complex systems in computational social science (Conte and Paolucci, 2014), land-use science (Matthews et al., 2007), political science (de Marchi and Page, 2014), computational economics (Tesfatsion, 2006), for the study of social-ecological systems (Schlüter et al., 2012; An, 2012), as well as in ecology (where it is often called individual-based modeling, Grimm and Railsback, 2005).[12]

Agents in ABMs can be individuals, households, firms or other collective actors as well as elements of the biophysical environment, for example fish populations. Agent behavior can be modeled at the individual level with any of the approaches introduced in Section 3 or other theories that can be formalized in equations, decision trees or rules. Agents are diverse, i.e. representing different types of agents that are characterized by specific attributes and decision making models (e.g., large and commercial versus small and traditional farms). Agents within a type are often also quantitatively heterogeneous in that they possess varying values of these attributes (e.g. regarding market access, social or financial capital). Agents interact on structures such as networks and their behavior is interdependent. They can adapt their behavior and learn. Together these characteristics can give rise to complex, often unpredictable aggregate behavior, patterns or functions (Page, 2015).

Because ABMs integrate individual decision making, heterogeneity and interactions between agents as well as between social and environmental processes, they are particularly suitable to study social-ecological systems as *complex adaptive systems* (Levin, 1998; Miller and Page, 2007), which are characterized by self-organization, adaptation, non-linear behavior and cross-scale emergence. *Self-organization* refers to the lack of a central control and the path-dependent evolution of patterns within the model (e.g. of groups of similar agents) through micro-level interactions over time. The system evolves through adaptations of heterogeneous and diverse agents and their behavioral strategies to the endogenously changing conditions of their social and ecological environment. ABMs therefore allow exploring non-linear behavior at the system level that emerges from interactions of micro-level structures and studying unexpected outcomes of micro- or macro-level disturbances or in-

---

[12]Note that in some scientific communities, this class of modeling approaches is also known as multi-agent simulations (MAS, Bousquet and Le Page, 2004).





terventions in the system. In this way, ABM helps to develop a mechanism-based understanding of system-level phenomena (Epstein, 1999; Hedström and Ylikoski, 2010). However, because of their potentially high complexity and dimensionality in state- and parameter space, ABMs are often difficult to analyze and may require high computational capacities to understand their dynamics beyond single trajectories.

In addition to the behavior of the agents, ABMs of social-ecological systems incorporate the dynamics of the environment resulting from natural processes and human action insofar as it is relevant for the agents' behavior or for answering a research question about its environmental and social consequences. For example, the decision to intensively use a land plot as pasture may lead to overgrazing and change the nutrient content of soils. This can ultimately make the land unusable and may force the agent to adapt a new strategy. Most ABMs of social-ecological systems describe agents as boundedly rational decision

makers (see Section 3.2) or profit maximizers that take into account information from the environment and other agents or social learners that imitate other agents (see Section 4).

In ABMs that describe systems at the local and regional level, agent behavior is often modeled through empirically-based agent types (Smajgl and Barreteau, 2014) or described by decision heuristics based on empirical observations of human behavior in specific situations (Conte and Paolucci, 2014). Furthermore, stochastic processes are used to capture uncertainty in and

the impact of random events on human decision making and assess the consequences for macro-level outcomes. For example, random events at the local level such as a random strategy change by an individual or environmental variation can give rise to non-linear macro-dynamics such as a sudden shift into a different system state (Schlüter et al., 2016).

In the context of land-use science, ABMs are mostly developed for local or regional study areas, taking into account local specificities and fitting behavioral patterns to data acquired in the field (Parker et al., 2003, 2008; Matthews et al., 2007). They

are often combined with cellular automaton models that describe the dynamics and state of the physical land system (e.g., Heckbert, 2013). In these ABMs, the spatial embedding of agents usually plays an important role (Stanilov, 2012).

The following example from the literature illustrates the ABM approach in the context of land-use science: Martin et al. (2016) model a number of cattle ranchers on a landscape that have to decide how to move their livestock on grassland patches. The patches are described by equations for biomass regrowth depending on the precipitation in the area. Therefore, overgrazing

in one year decreases feed availability in the following year. Agents decide when, where and how many cattle to graze on a particular land patch based on their individual goals or needs, information on the state of the grassland, beliefs about the future and interactions with other ranchers. Such a model can reveal the interplay of different land-use strategies on common land and help to assess the vulnerability of land-use strategies to shocks such as droughts.

Agent-based approaches can be applied without modeling each individual agent explicitly. It suffices to model a representa-

tive statistical sample of agents that depict the important heterogeneities of the underlying population. To capture major types of human behavior, a recent proposal are *agent functional types*[13] based on a typology of agent attributes, interactions and roles (Arneth et al., 2014). This proposal is explored for modeling the adaptation of land-use practices to climate change impacts (Murray-Rust et al., 2014). Agent-functional types represent a typology that is theoretically constructed instead of an empirically derived data-driven typology, which is common in empirically-based ABMs. Such agent-based approaches are promising

---

[13]Arneth et al. (2014) make the analogy to plant functional types in vegetation models.




for Earth system modeling with respect to addressing questions of interactions across levels, for instance how regional or global patterns of land use emerge from local or regional land-use decisions that are the results of local interactions at the respective level.

### 5.6 Dynamics at the system level: System dynamics, stock-flow consistent and input-output models

This final subsection discusses modeling approaches without explicit micro-foundations. Decisions in such models are not modeled directly but, as policy decisions in integrated assessment models, through the construction of different scenarios for the evolution of crucial exogenous parameters in the model. Because the dynamics are not explained by decisions of individual agents, such approaches deviate from the standards of methodological individualism.

*System dynamics* models describe the economy, population and crucial parts of the Earth system as well as their dynamic
interactions at the level of aggregate dynamic variables, usually modeling the dynamics as ordinary differential equations or difference equations to map future developments. The equations are often built on stylized facts about the dynamics of the underlying subsystems and are linked by functions with typically many parameters. Modelers employ systems dynamics models to develop scenarios based on different sets of model parameters and assess system stability and transient dynamics of a system. In comparison to equilibrium approaches, systems dynamics models capture the inertia of socio-economic systems
at the cost of a higher dimensional parameter space. This can lead to more complex, e.g. oscillatory or overshooting, dynamics. Systems dynamics models can be very detailed, like the World3 model commissioned by the Club of Rome for their famous report on "Limits to Growth" (Meadows et al., 1972, 2004), the GUMBO model (Boumans et al., 2002), or the International Futures model (Hughes, 1999). Subsystems of such models comprise human population (sometimes disaggregated between regions and age groups), the agricultural and industrial sector, as well as the state of the environment (e.g. pollution and resource
availability). Simpler models describe the dynamics of only a few aggregated variables at the global level (Kellie-Smith and Cox, 2011) or confined to a region (Brander and Taylor, 1998).

System-level approaches to macroeconomic modeling often emphasize self-reinforcing processes in the economy and point at positive feedback mechanisms, resulting in multi-stability or even instability (e.g., increasing returns to scale in capital accumulation and self-amplification of expectations during economic bubbles). For example, post-Keynesian and ecological
economists use stock-flow consistent models to describe a circular economy in which low aggregate demand can lead to underutilization of production factors and the state plays an active role to stabilize the economy (Godley and Lavoie, 2007). In these models, a social accounting matrix provides a detailed framework of transactions (flows, i.e. per-time quantities) between representative agents in the economy such as households, firms and the government, which hold stocks of financial and physical assets and commodities (Godley and Lavoie, 2007).

While stock-flow models often focus on the monetary dimension of capital and goods, ecological modeling approaches focus on material accounting or try to integrate material with financial stocks and flows in one framework (e.g., Berg et al., 2015).*Input-output modeling* focuses on the material side of economic production (Leontief, 1986; Ten Raa, 2005; Miller and Blair, 2009) and can be extended to analyze the industrial metabolism, i.e. the material and energy flows and its environmental impacts in modern economies (Fischer-Kowalski and Haberl, 1997; Ayres and Ayres, 2002; Suh, 2009). Input-output models





consider different sectors or production sites of the economy and the material inputs that are needed to produce a desired good together with unwanted side-products such as waste and pollution. Each sector or production site is modeled by a fixed proportions ("Leontief") production function, which is characterized by linear factors that depend on the available technology. Given a final demand, the required production of all intermediate goods and the resulting environmental footprint can be

calculated by linear programming techniques. Regional input-output models also account for spatial heterogeneity and are used for example to estimate the environmental footprints of industrialized countries in other regions (Wiedmann, 2009) or to evaluate possible impacts of extreme climate events on the global supply chain (Bierkandt et al., 2014).

In the context of land-use change, an input-output model could describe which primary input factors such as land, fertilizer, machinery, irrigation water and labor are required for satisfying the demand of an agricultural commodity by a specific pro-

duction technique. Some of these primary inputs have to be produced themselves, using other inputs. Outputs may also include side-products such as manure in cattle production or externalities such as environmental degradation. The model could be used to compare different technologies or explore how changes in demand would lead to higher-order effects along the supply chain, for instance differences between intensive agriculture (mono-cultures for animal feed) and extensive land-use such as cattle ranching.

In order to make these techniques useful for modeling the impact of humans on the Earth system, they could be combined with approaches that model the development of new production technologies and how they are affected by decisions at different levels (consumers to policy makers).

## 6   Discussion

In the previous three sections, we reviewed different approaches to model individual human decision making and behavior,

interactions between agents and to aggregate these processes. We illustrated them with examples from the land-use context and discussed their potential application in Earth system modeling with the aim of modeling complex feedback dynamics between natural and social components.

One intention of this review is to draw the attention of the reader to the different assumptions and theories about human decision making and behavior that are possible to describe with specific modeling approaches and point to their strengths and

limitations. Some modeling techniques are compatible with almost any theory of human behavior or decision making that can be formalized and can thus be used with many assumptions about human behavior. Other modeling techniques significantly constrain possible assumptions about human behavior and decision making. Therefore it is important to first decide on which assumptions about human behavior are reasonable in the context of a research question and then choose the techniques accordingly. To put it the other way around, the choice of a specific modeling technique may have considerable consequences

for the types of meaningfully answerable research questions and kinds of analysis that they can provide. Modeling choices require a constant interplay between model development and the research questions that drive it. In Table 5, we summarized the approaches we discussed in this paper and collected important questions regarding the different categories. We think a





**Table 4.** Summary table for aggregation and system level descriptions

| Approaches and frameworks | Key considerations | Strengths | Limitations |
|---|---|---|---|
| Social utility and welfare: Aggregate individual utility, possibly taking inequalities into account | How is inequality evaluated? How is welfare compared between societies and generations? | Base for cost-benefit analysis, a widely applied decision model for policy evaluation | Assumes that individual utility can be compared on a common scale |
| Aggregation via markets: Representative agents in economic models | Which preferences do representative agents have? Which spatial and temporal scales do price mechanisms span? | Well developed formalism that makes the connection between micro- and macroeconomics analytically traceable | Assumes that aggregated agent properties are similar to individual ones to derive economic equilibrium, coordination effort between agents neglected |
| Social planner and economic policy in integrated assessment models: Model possibilities to internalize environmental externalities | Which economic policy instruments internalize environmental externalities best? What are plausible scenarios for policy implementation? How do agents react to changes in policy? | Allows to determine optimal paths for reaching societal goals | Models focus on production and investment in the economy |
| Distributions and moments: Model heterogeneous agent attributes via statistical properties of distributions | Which heterogeneities are most important for the macro-outcome? | Systematic way to analytically treat heterogeneities | Only applicable for rather simple behaviors and interactions |
| Agent-based models: Simulate agent behavior and interactions explicitly to study emergent macro-dynamics computationally | Which kind of agents types are important? How do the agents interact with each other and the environment? | Very flexible framework regarding assumptions about decision rules and interactions | Models often with many unknown parameters, difficult to analyze mathematically |
| Dynamics at the system level | Which crucial parameters in the model can be influenced by decision makers? | Allows to explore possible dynamical properties of the system based on macro-mechanisms | No explicit micro-foundation |





modeler who needs to make decisions on which approaches and techniques to use in order to include humans into Earth system models should be aware of the questions and considerations, which we discuss in the following.

Regarding individual agents, we identified three important determinants in decision models: motives, restrictions and decision rules. Assumptions about each of these three determinants need to be made with great care, as there are many factors that

might influence which motives, restrictions, and decision rules are relevant in a given context. For instance, modelers make different assumptions about whether decision makers only consider financial incentives or whether also soft incentives, such as a desire for fair outcome distributions (Fehr and Gächter, 2002), are relevant (Opp, 1999). Research shows that the relevance of motives can vary over time and that surprisingly subtle cues can change the importance of motives (Lindenberg, 1990; Tversky and Kahneman, 1985). Likewise, the choice of a plausible decision rule depends on the studied context. For instance, a decision

rule that requires complex calculations may be relatively plausible in contexts where individuals make decisions with important consequences and where they have the information and time needed to compare alternatives. When stakes are low and time to decide is limited, however, more simple decision rules are certainly more plausible. Cognitively demanding decision rules are also more plausible when decision makers are collectives, such as companies and governments. Sometimes, it may even be reasonable to assume that actors use combinations of the different decision models (Camerer and Hua Ho, 1999).

When focusing on the interaction of agents, important criteria for choosing an appropriate model are the type and setting of interactions, the assumptions that agents make about each other, the influence they may exert on each other and structure of interactions. For example, interactions in competitive environments will only lead to cooperation if this is individually beneficial. In such environments, agents may assume that the others' form their strategies rationally. In less competitive settings, where social norms and traditions play a crucial role, however, behavior may not be strategically chosen but rather adaptively,

e.g., by imitating other agents. This might also be important on time scales at which cultural evolution happens. Furthermore, social settings might favor that agents influence each others' characteristics and primarily interact by exchanging opinions or sharing beliefs.

Finally, an important criterion for the choice of how to model interactions of single elements is whether the local structure of interaction matters. If it does, this would require a gridded or networked approach, otherwise a mean field approximation is

justified. Similar choices have to be made in classical Earth system models: For example, the interaction of ocean and atmosphere temperature near the surface on a spatial grid could be modeled either by only taking interactions between neighboring grid points into account or by coupling the ocean temperature to the atmospheric mean field. Analogously, the interactions between groups of two types of agents may be modeled explicitly on a social network. However, it might also suffice to only consider interactions between two agents representing the mean of each group respectively. The question whether the interac-

tion structure matters can often not be answered a priori but can be the result of a comparison between an approximation and an explicit simulation.

For the aggregation of individual decision making and interactions, crucial criteria for modeling choices are the properties of relevant societal aggregation mechanisms, decision criteria for collective agents, heterogeneity of modeled agents and relevant time and spatial scales of macro-descriptions. We introduced different approaches to model political and economic institutions

(markets, voting protocols) that aggregate individual decisions. Depending on the specific research questions, such modeling




approaches can be adopted to fit particular real-world system and describe their aggregation. To model decisions between economic or environmental policies, normative decision models can sometimes also be used to describe such decisions if they take into account actual and perceived controls of policy makers and consider the effect of compromises between different interest groups.

Furthermore, there are interesting parallels in choices of modeling techniques between classical Earth system modeling and socio-economic models at the macro-level. We discuss here two examples: First, the choice between different aggregation techniques to connect a micro- to the system-level depends on the expected importance of interactions and heterogeneity in an assumed set of agents. Take as an example vegetation models, in which modelers may consider representative plant functional types or individual adaptive plants depending on whether they consider the interaction and heterogeneity important for the

macro-dynamics. Analogously, a model of social dynamics may choose for instance between a representative agent or model heterogeneous agents explicitly in an agent-based model. Of course the choice between a coarse-grained and a fine-grained description crucially depends on the properties of the system and the research questions.

Second, time scales help to decide whether elements and processes should be modeled as evolving in time, fixed or need not be considered at all for a macro-level description of the system. As an example, consider the propagation of increased $CO_2$

concentration in global circulation models. The relatively quick convection of $CO_2$ in the atmosphere may not be of interest on longer time scales and the $CO_2$ concentration can be assumed to be well-mixed. But when modeling $CO_2$ concentrations in the oceans on politically relevant time scales, the assumption that $CO_2$ is well-mixed might distort the results considerably because convection between ocean layers is comparatively slow (Mathesius et al., 2015). Similarly, general equilibrium models can be a good description if the convergence of prices happens on fast time scales and market imperfections are negligible.

Dynamical systems models, on the contrary are more appropriate to describe systems with a high inertia that may operate far from equilibrium due to continuous changes in system parameters and slow convergence. Questions about the relevant spatial scales and the importance of the location of entities can be similarly related to modeling decisions in classical Earth system models.

As we have shown, there are some similarities regarding the choice of modeling techniques and assumptions in Earth system

models and models of socio-economic systems. However, fundamental differences between the modeled systems pose a big challenge for an informed choice of modeling techniques. Earth system models can often build on fundamental scientific laws describing micro-interactions that can be tested and scrutinized. Of course this can result in very complex macroscopic system behavior with high uncertainties. But models including human behavior on the other hand have to draw on a variety of accounts of basic motivations in of human decision making. And these motivations may change over time while societies evolve and

humans change their actions because of new available knowledge.

At this point, there is a crucial feedback between the real world and models: Agents (e.g. policy makers) may decide differently when they take the information provided by model projections into account. Therefore, it is important to keep in mind that modeling activities including humans might eventually change the behavior of agents because of human reflexivity. This makes models of human-dominated systems fundamentally different from natural science models. This also points to the

difference in social modeling between normative and descriptive model purposes. We highlighted this difference throughout



the paper but want to point out its importance here again: The purpose of a model is crucial for the choice of suitable modeling techniques, e.g. choosing a maximization technique or a set of differential equations to tackle the research question. Or to put it the other way around: The choice of a modeling approach may already imply basic normative assumptions although the modeler may not be aware of them and just choose it for pragmatic reasons.

A difficulty that we encountered in the classification and presentation of the material is that it is not always clear which parts of a theory are important assumptions regarding human behavior and which modeling decisions are taken because of mathematical convenience. Choosing assumptions for technical reasons, e.g., mathematical simplicity and tractability, may be problematic because it remains unexplained how they are related to the real world. Additionally, assumptions in one model may be understood as descriptive (positive) statements whereas in another model they may be meant as prescriptive (normative)

ones, depending on the application of the model. For example, in a model of agricultural markets, the assumption that big commercial farms maximize their profits might be a reasonable descriptive approximation. However, in a model that asks how small-holder farms could survive under such market conditions, the same assumption gets a strong normative content.

Another important insight from the reviewing effort is that the terminology sometimes differs between disciplinary or sub-disciplinary scientific fields. Therefore, different terms from two separate fields could refer to very similar theories whereas the

same term might be used to describe quite separate varieties of a theory in different fields. We tried to use our terminology as clearly as possible and thus hope to contribute to a better understanding between different fields. But not only the terminology differs between fields, there are also important differences in focus and consequent limitations between different schools in the social sciences.

We also want to point out that our survey of techniques has a bias towards economic modeling techniques for two simple

reasons: First, economics is the social science discipline that has the longest and strongest tradition in formal modeling of human decision making. Second, economics focuses on the study of production and consumption as well as the allocation of scarce resources. In most industrialized countries today, a major part of human interactions with the environment is mediated through markets, central in economic analyses. This review goes beyond the often narrow framing of economic approaches while at the same time not ignoring important economic insights. For instance, consumption and production decisions do not

only follow purely economic calculations but are deeply influenced for instance by behavioral patterns, traditions and social norms (The World Bank, 2015).

However, many theories in the social sciences are building on verbal models rather than mathematical formalizations. A major part of the theoretical work in the social sciences is very context specific and some sub-disciplines reject that their findings can be meaningfully generalized (Williams, 2012; Rosenberg, 2012). However, including human decisions into Earth

system models requires such generalizations. Without them, the modeling of futures that are potentially very different from the past would not be possible. This makes many approaches in the social sciences incompatible with natural science and therefore difficult to include into Earth system models. It is therefore important to put effort into formalizing such theories, making them scalable and testing the consequences of their different assumptions about human behavior, interactions and its aggregation at different levels.





**Table 5.** Collection of questions that may guide the choice of modeling approaches and assumptions.

| Category | Important modeling questions |
|---|---|
| Modeling individual decision making and behavior | Which goals do agents pursue? |
| | Which constraints do they have? |
| | Which decision rules are used? |
| | How do agents acquire information and beliefs about their environment? |
| Modeling interactions between agents | Do agents interact in a competitive environment or are interactions primarily governed by social norms? |
| | What do agents assume about each other's rationality? |
| | Do agents choose actions strategically or adaptively? |
| | How are agents influenced by others regarding their beliefs and norms? |
| | Which structure do the interactions have? |
| Aggregating behavior and modeling dynamics at the system level | Are agent decisions aggregated through political institutions (e.g., voting procedures) or markets? |
| | According to which criteria do policy makers decide and which controls do they have? |
| | Is the heterogeneity of agent characteristics and interactions important? |
| | Which macro-level measures are dynamic and which can be assumed to be fixed? |

Most global models that describe human interactions with the Earth system we found in the literature are based on economic assumptions about the behavior of societies and are often only linked in a one-way fashion to the biogeophysical part of the Earth system. It is thus an ongoing challenge to include co-evolutionary dynamical interactions of human societies with other Earth system components into global models. Of course, models including human decision making and social dynamics can

5   not claim to describe all real-world social interactions. However, they could include formal descriptions of idealized social mechanisms so far not considered explicitly in global models that are regarded as being important to explain driving forces for environmental impacts such as land-use change. Even though more realistic models would have to be much more complex if they considered the heterogeneity of agents in all relevant aspects, in many real-life settings even simple conceptual models of social mechanisms are good descriptions of key features of the dynamics at work. One the one hand, modeling human

10   behavior comes with many degrees of freedom. On the other hand, modelers need to choose assumptions that are plausible in the context of their study, consulting existing social-scientific research, exploring whether alternative assumptions about the three determinants are crucial in the sense that they affect the predictions of the ESM, and conducting the empirical research needed to select the appropriate model.



## 7 Summary and Conclusion

In this review, we discussed common modeling techniques and theories that could be potentially used to include human decision making and the resulting responses into Earth system models. Although we could only discuss basic aspects of the presented modeling techniques, it is apparent that modelers who want to include humans into Earth system models are confronted with crucial choices of which assumptions to make about human behavior and which appropriate techniques to use.

As Table 5 summarizes, we discussed techniques and modeling assumptions in three different categories. First, the modeling of individual decision making focuses on decision processes and the resulting behavior of single agents and therefore has to make assumptions about how choices between possible behavior comes about. Second, models of interactions of two or several agents capture how decisions depend upon each other and agents influence are influenced regarding different decision criteria. Third, modeling techniques that aggregate individual behavior and interactions to a system level description. The third category is crucial for being able to model human behavior at scales relevant for the Earth system but requires key elements of the first and the second categories. Finally, we discussed important questions regarding the choice of modeling approaches and their interrelation with assumptions about human behavior and decision making, e.g. regarding the level of description, the relevant time scales but also difficulties that can arise due to human reflexivity and the amalgamation of normative and descriptive assumptions.

The formal models used in various disciplines to describe human behavior in environmental contexts have a bias toward economic approaches. This is not surprising because most of the modeling techniques applied in the context of global environmental change today follow economic approaches and many human interactions with the environment are driven by economic forces. However, we think that it is necessary to advance research that also applies social modeling and simulation in the context of global environmental change. One important aim of such research would be to provide a theoretical basis for including processes of social evolution and institutional development into Earth system models. If we want to explore the possible futures of the Earth, we need to get a better understanding of how the long-term dynamics of the Earth system in the Anthropocene is shaped by these social processes.

A new generation of Earth system models can build on various approaches, some of which we reviewed here, to include human decision making and behavior explicitly into Earth system dynamics. However, ambitious endeavors like this have to take into account that modeling of human behavior and social processes is a contested topic and the assumptions and corresponding modeling techniques need to be chosen carefully being aware of their strengths and limitations for the specific modeling purpose.

*Competing interests.* The authors declare that they have no conflict of interest.

*Acknowledgements.* The idea for this paper emerged during discussions at the LOOPS workshop 2014. We thank all the participants for their contributions. Furthermore, FMH thanks Silvana Tiedemann, Tim Kittel and Felix Kersting for their comments on previous versions of the paper. FMH acknowledges funding by the DFG (IRTG 1740/TRP 2011/50151-0). MS acknowledges funding by the European Research



Council under the European Union's Seventh Framework Programme (FP/2007-2013)/ERC grant agreement no. 283950 SES-LINK and a core grant to the Stockholm Resilience Centre by Mistra. JFD thanks the Stordalen foundation (via the Planetary Boundary Research Network PB.net), the Earth Doc network and the Leibniz Association (project DOMINOES) for financial support.



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
