# Peer review of "Towards representing human behavior and decision making in Earth system models – an overview of techniques and approaches"

_Earth System Dynamics, 2017_

## Referee Comment (RC2) · Anonymous Referee #2 · 5 Jun 2017

[Posted by editor on behalf of Reviewer 2, review received 31 March 2017]

This paper provides a very comprehensive review of the application of human behavior in earth system models. I was impressed with the coverage and extensive literature review. The paper is well written and will make a valuable contribution to the field. My main concern, which perhaps is unavoidable for such a review, is that the paper is very long, bordering on overwhelming. There are parts that are redundant such as page six, which takes three paragraphs to restate a Table. I suggest the authors search for other places to streamline the paper. The table in the Discussion is an excellent summary. I would recommend publication following minor revision.

---

## Editor Comment (EC1) · S. Cornell (Editor) · 6 Jun 2017

Both reviewers agree that the paper gives a good overview and a useful structuring of the wide range of approaches to representing behaviours and decisions in ESMs.

Reviewer RC1 gives very constructive guidance for developments to the ideas of this paper. I propose that the author team deals briefly with points 2-5 in their revisions to the current paper, and then takes up the reviewer's other suggestions in a follow-up paper. The current overview is useful in itself, but setting out guiding principles of model framework selection (providing a detailed exploration of "when is ESM-peopling useful?"), with a hard-hitting critique of the approaches involved, would be a really

valuable contribution to transdisciplinary ESD debates.

---

## Author Response (AR1)

**Referee Comment #1**

We thank the referee for his valuable comments. Although we do not agree with all the points, we think that they raise important issues that could be clarified in the paper. Furthermore, a productive ongoing discussion about these issues could help in aligning forces for the important goal of gaining a more holistic understanding of global human-nature interactions by developing Earth system models that include important social and economic dynamics. In the following, we respond point by point to the comments of the referee.

> *This paper provides an overview of a broad range of representations of human behaviours that might be considered when attempting to 'people' Earth System Models (ESMs). I found the paper to be well researched and written on the whole and if the aim was to inform the reader as to the range of options on offer in this space it did a relatively good job (with one or two notable exceptions which I detail below). However, the title suggests something more, with the stated aim to also offer some guidance over the way forward in this space. This is very much needed given the likely expansion of research this area will experience. Unfortunately, I found this aspect of the paper a*
> *little disappointing given it was rather passive, reserved or limited in any guidance it offered. This was not helped by the structure of the paper which separated out the extensive review of potential methods and the critique of these methods which was largely relegated to the Discussion. If the authors really want to be faithful to their title and stated aims I would suggest some editorial changes. I would start by offering a strong steer on the guiding principles of model framework selection in this space. I would then combine the description of the options with a more hard-hitting critique of the various options assessed against your guiding principles. My reading of the current*
> *paper suggest the author team would be more than able to achieve this and the product would be far more valuable than the largely descriptive review currently tabled. The alternative would be to dilute the title and aims to being those of a review of options as I believe this is what is currently being offered. I would like to encourage the former but providing the title and aims were adjusted the paper could go forward without this reediting. I've ticked the 'major revisions' box but only because I couldn't simultaneously tick the 'minor revisions' box. This depends on which way you chose to jump.*

We appreciate the critique of the referee and agree that this work did not deliver on the promise of a general guideline for building ESMs with explicit human decision and behavior components. This is for a specific reason: Such a guideline depends a lot on the concrete research questions that a modeler wants to tackle with the model. Therefore we argue that rather than a concrete guideline, some general principles have to be considered by the modelers and they have to be aware of the various possibilities from the toolbox that the literature provides and we aim to give an overview over. This approach is much in line what researchers from sociology have termed theory of the middle range (Merton, 1957). This approach does not aim at an all-encompassing theory of whole societies, but rather argues for using elements of different theories tailored to a specific problem. The selection of assumptions underlying the modeling approach has to be on the ground of good reasons and empirical evidence. In case of doubt, the validity of assumptions have to be tested for the specific context. Furthermore, we note that an extensive critique of all the different methods would be beyond the scope of a single paper. Where we were aware of such critiques, we provided some references for the readers. However, due to the huge variety of methods, there may be relevant strands of critique which we were not aware of and therefore did not include into the paper.
In line with the above considerations, we will change the title and make the aim of the paper clearer in the introduction to avoid misunderstandings. Furthermore, we will make the general point more

prominent, that there is not one method and theory that will fit all relevant research questions, which are interesting in the context of global human-nature interactions. Therefore the approach most appropriate for the question at hand has to be selected taking into account various general considerations as listed in the Discussion part of the paper.

> *Specific points (in no particular order)*
> *1. I would like to see a full discussion over when ESM peopling might be useful, when it might not and when it might be actively discouraged. Given the huge uncertainties this activity can/will open up researchers need dissuading from the illegitimate and unnecessary hybridisation of social and natural systems models. This paper could offer some guiding principles. For example, although the chosen example of land surface/use parameterisation suggest a useful role for microscopic representations of people, ultimately we are only interested in the structural social dynamics when exploring Earth (i.e. global) scale feedbacks, even if these dynamics arise from the act of an individual. Therefore, at the ESM scale you would have to have a really powerful justification of a highly disaggregated representation of people and there should always be a presumption in favour of the macroscopic representation. The fact that ESMs are spatially disaggregated and therefore we should naturally entertain representations of people at this scale is not sufficient in my view.*

We agree with the referee that a discussion about when a "peopling" of ESMs is useful should be added to the paper. We will add some corresponding paragraphs to the paper discussing that this is only relevant if there is a closed loop of interactions with the outcome of relevant decision processes and behaviors changing over the relevant time scales. However, we think that a full-blown discussion of this question could be well suited for a follow-up paper as suggested by the editor.

Regarding the example of the macro- vs. micro-description of a human component in ESMs, we want to note that we do not argue that human behavior always has to be included at a micro-level and on the basis of single actors. But, as we are arguing in the paper, a complete picture of humans in ESMs should be well founded in micro-models of decision making, behavior and interaction. Especially when large societal and institutional changes are considered, models purely based on observed macro-dynamics might not be able to rightly capture these changes (this is referred to as the Lucas-critique in the economic literature). Of course, here again, it depends on the research questions whether a macro model of societal dynamics suffices (assuming that major societal dynamics will not change fundamentally over time) or if a more micro-founded model is needed.

> *2. The opening text made a big play of the distinction between 'explicit decisions' and 'implicit behaviours'. Close inspection suggests this is a largely arbitrary distinction and some critique of this divide would be a useful addition. Is me typing this response an explicit decision or an implicit behaviour? I'm not sure.*

If the question is based on the reading of our definition that decisions are only explicit and there are no implicit processes involved, then we regret the misunderstanding. We reformulated the corresponding paragraph to make it clear that decision-making can be influenced by implicit, unconscious and intuitive processes. In this understanding, the result of a decision process is usually a certain type of behavior.

However, not every behavior has to be the outcome of a decision process, and this is why we have to insist that the distinction between decision making and behavior is analytically useful and not arbitrary. Although in the end, only the behavior of humans may be observable, many behaviors are highly influenced by semantic considerations as well as inscribed social and individual norms and values. For complex cultural settings, it is therefore often not helpful to reduce humans to a reflex-response scheme as in behaviorist approaches.

The only alternative to modeling behavior without explicitly using theory about the decision processes would be to model behavior statistically or at the basis of physiological processes in the brain. Concerning the latter, the science is still in its infancy and it is at least questionable whether such a description is possible at all. Regarding statistical approaches, as explained in the previous point, when looking at strong social changes, statistical correlations might break down calling for the explicit modeling of decision processes.

Apart from these more pragmatic considerations, there is a philosophical argument to be made: From introspection the distinction between behavior as an event of the physical world (i.e., the body) and the decision-making process as at least being influenced by the mind should be clear to every human making conscious decisions. How these different processes interact has been the subject of the age-old debate called the mind-body problem in philosophy. Solving this problem by simply denying the existence of the mind altogether leads to even more serious problems: If we would assume that me typing this response is only a behavioral reaction to a very complex stimulus without any involvement of semantic processing, why should anybody of us care about the semantic content of want we are writing here anyways?

> *3. Surely the most important distinction in normative framing involving any ESM is whether they adhere to the current socio-economic norm or they represent transitional/ transformative dynamics. Everything else is simply detail. This is not developed at all and yet practically all applications of peopled ESMs will revolve around exploring and contrasting alternatives to business-as-usual. This review is very constrained in this regard, and hardly mentions alternative (and potentially indispensable) economic framings required when investigating, for example, implementation of the Paris Agreement.*

We are well aware of the debate between the economic mainstream dominated by neoclassical theory and heterodox schools of economic thought and the different economic framings they involve (see publications of the lead author). To come up with new models of the economy that build on the work done in heterodox branches such as ecological and institutional economics is actually one of the main challenges when building social dynamics into ESMs. Thus, we agree that such models have to go beyond the currently dominant socio-economic framing. However, we tried to avoid an extensive discussion of this debate in the paper. The main goal of this paper is to compare different approaches to modeling human decision making that could be potentially useful to Earth system modeling. Therefore, the paper only considers those economic approaches that use mathematical modeling. Because many of the heterodox economic schools are not much engaged with modeling or event reject mathematical modeling as a valid tool to advance knowledge about social processes, this collection, unfortunately, is much biased towards mainstream economic thinking. If we omitted important and formalized economic modeling approaches in the literature, this is only due to our limited knowledge.

> *4. Other than discussion of flow consistent approaches, this review makes little or no mention of (bio)physical frameworks as covered in say ecological economics. I appreciate they are not mainstream but I think this is a critical omission because perhaps the most consistent scheme for peopling of ESMs is where both the Earth and social systems are both on a sympathetic '(bio)physical' footing. This could be nicely contrasted against the fact that the standard macroeconomic framings are flow/physically inconsistent. Perhaps it's time for the natural sciences to call the macroeconomic emperor on their lack of physically defensible clothing and peopling ESMs appears to be a great place to start. ESD has been central to getting these alternatives into the literature and it is anomalous that they are not considered here.*

A discussion of purely biophysical models is neither the goal nor the focus of our article. We agree that a biophysical description of human activities is crucial for linking classical ESMs and social

science approaches and that physically consistent stock-flow or similar models should be an essential part of ESMs with explicit human dynamics. Therefore, we will improve our account of physical stock-flow consistent modeling and add references to the important work of Nicholas Georgescu-Roegen in this area. We also agree that models of the social metabolism have to take thermodynamic limits into account. However, we doubt that thermodynamic laws alone can account for the complex dynamics of social-metabolic processes as some recent work of the referee and others in this special issue suggest (Garrett, 2014; Garrett, 2015; Jarvis et al., 2016).

> *5. Much of the problem space that peopled ESMs would explore would be around precautionary Command and Control type policy such as that offered in the Paris Agreement. Here a formal control representation of 'people' is much more appropriate given it is about compliance or non-compliance with a stated environmental objective such as keeping below 2 K. I would like to see some discussion of this.*

Actually, a lot of economic reasoning for environmental policy recommendations builds strongly on the control perspective. But as the failure of some of these policies shows, it is not only important to have the formal framework right but also the micro-model of human behavior and decision making to judge how people will react to changes in institutional frameworks. For example, in some settings monetary incentives for environmental behavior might be counterproductive because they can lead to crowding out effects when moral rules are replaced by economic considerations. Therefore, a successful policy assessment needs to select correct micro-models to identify the right approaches for adjustments that influences individual behavior in the right direction. This applies equally to command and control type policies as to other (e.g., market-based) solutions. As suggested by the referee we will add these considerations to the paper.

**Referee Comment #2**

*This paper provides a very comprehensive review of the application of human behavior in earth system models. I was impressed with the coverage and extensive literature review. The paper is well written and will make a valuable contribution to the field. My main concern, which perhaps is unavoidable for such a review, is that the paper is very long, bordering on overwhelming. There are parts that are redundant such as page six, which takes three paragraphs to restate a Table. I suggest the authors search for other places to streamline the paper. The table in the Discussion is an excellent summary. I would recommend publication following minor revision.*

We thank the referee for the positive response. We will revise the paper, shorten the suggested parts, and aim at an overall reduction of the text.

**List of changes in the manuscript**

Important note: The co-author Rainer Hegselmann withdraw his authorship because he was not able to take part in the revision process and held that his contribution to the paper would not justify a co-authorship.

We included all the changes as promised in the above response to the reviewers comments. These include:
- We changed the title.
- We changed the introduction, discussion and abstract to make the goals of the paper clearer.
- We included a part in the introduction explaining that there is no single approach/technique usable for all relevant questions regarding human-nature interactions on a global scale and that researchers have to choose the techniques to model decision making and behavior appropriate for the specific context.
- We added some brief discussion of the question when modeling of human decision making explicitly is useful.
- We extended the discussion of models from the field of ecological economics.
- We made it clear in the introduction that a micro-based approach to human behavior is important for exploring the impact of environmental policies (even from a control-theory perspective).
- We shortened the paper as much as possible, given the requests by the first referee to extend some parts of the introduction, discussion and some subsections of the main parts.

Furthermore, we made the following changes:
- We moved the discussion of the methodological question how to model social systems (methodological individualism vs. structuralist approaches) from the introduction to the second section because it fits there much better.
- We corrected typos in the former version of the paper, changed wording that caused misunderstanding with some readers of the discussion paper and updated the references. A detailed comparison of the original submission and the resubmitted version of the paper can be found below.

```
%% Copernicus Publications Manuscript Preparation Template for LaTeX Submissions
%% ---------------------------------
%% This template should be used for copernicus.cls
%% The class file and some style files are bundled in the Copernicus Latex
Package, which can be downloaded from the different journal webpages.
%% For further assistance please contact Copernicus Publications at:
production@copernicus.org
%% http://publications.copernicus.org/for_authors/manuscript_preparation.html

%% 2-column papers and discussion papers
\documentclass[esd]{copernicus}

%% \usepackage commands included in the copernicus.cls:
%\usepackage[german, english]{babel}
%\usepackage{tabularx}
%\usepackage{cancel}
%\usepackage{multirow}
%\usepackage{supertabular}
%\usepackage{algorithmic}
%\usepackage{algorithm}
%\usepackage{amsthm}
%\usepackage{float}
%\usepackage{subfig}
%\usepackage{rotating}

\begin{document}

\title{Towards representing human behavior and decision making in Earth system
models – an overview of techniques and approaches}

\Author[1,2]{Finn}{M\"{u}ller-Hansen}
\Author[3]{Maja}{Schl\"{u}ter}
\Author[4]{Michael}{M\"{a}s}
\Author[5, 6]{Rainer}{Hegselmann}
\Author[1,3]{Jonathan F.}{Donges}
\Author[1,2]{Jakob J.}{Kolb}
\Author[1]{Kirsten}{Thonicke}
\Author[1]{Jobst}{Heitzig}

\affil[1]{Potsdam Institute for Climate Impact Research, Telegrafenberg A31,
14473 Potsdam, Germany}
\affil[2]{Department of Physics, Humboldt University Berlin, Newtonstra{\ss}e
15, 12489 Berlin, Germany}
\affil[3]{Stockholm Resilience Center, Stockholm University, Kr\"{a}ftriket 2B,
114 19 Stockholm, Sweden}
\affil[4]{Department of Sociology and ICS, University of Groningen, Grote
Rozenstraat 31, 9712 TG Groningen, The Netherlands}

\affil[5]{Frankfurt School of Finance \& Manangement, Sonnemannstraße 9-11
60314 Frankfurt am Main, Germany}
\affil[6]{Bayreuth Research Center for Modeling and Simulation, Bayreuth
University, Universit\"{a}tsstrasse 30, 95440 Bayreuth, Germany}

\runningtitle{Approaches to represent human behavior in ESM}

\runningauthor{M\"{u}ller-Hansen et al.}

\correspondence{Finn M\"{u}ller-Hansen (mhansen@pik-potsdam.de)}

\received{}
\pubdiscuss{} %% only important for two-stage journals
```

```latex
\revised{}
\accepted{}
\published{}

%% These dates will be inserted by Copernicus Publications during the
typesetting process.

\firstpage{1}

\maketitle
```

```latex

[revised manuscript text omitted]

    **\label{tab:levels}**
    \begin{tabular}{p{1.5cm} L{3cm} L{3cm} p{4cm} p{4cm}}
    \tophline
    Level & Socioeconomic units &
Fields/Communities & Common approaches and theories & Common assumptions about
decision making \\

    \middlehline
    \multirow{2}{1.5cm}{Micro} & Individual humans &
Psychology, neuroscience, sociology, economics, anthropology
    & Rational choice, bounded rationality, heuristics, learning theory,
cognitive architectures &
    [All assumptions presented in this column]
    \\
    &
Households, families, small businesses &
Economics, anthropology &
Rational choice, heuristics, social influence &
Maximization of consumption, leisure, profits
    \\

    \middlehline
    Intermediate &
Communities (villages, neighborhoods), cities &
Sociology, anthropology, urban studies &
Social influence, networks &
Transmission and evolution of cultural traits and traditions
    \\

    &
Political parties, NGOs, lobby organizations, educational institutions &
Political science, sociology &
Strategic decision making, public/social
choice, social influence and evolutionary interactions &
Agents form coalitions
 to achieve goals, influenced by beliefs and opinions of others

    \\
    & Governments &
Political science, operations research &
Strategic decision making, cost-benefit and welfare analysis, multi-criteria
decision making &
Agents choose for the common good
    \\
    & Nation states, societies &
Economics, political science, sociology & welfare maximization, social choice &
Majority vote
    \\

    \middlehline

```
Global & Multinational firms, trade networks &
Economics, management science &
Rational choice &
Maximization of profits or shareholder value
\\

& Intergovernmental organizations &
Political science (international relations)  &
Strategic decision making, cost-benefit analysis &
Coalition formation
\\

\bottomhline
\end{tabular}
\belowtable{} % Table Footnotes
\end{table*}

%
================================================================================
%
================================================================================
```

In the following three sections, we introduce the modeling techniques that are used in the literature to describe human behavior, interactions between individuals, and to aggregate them between the different levels. We start this overview at the level of individual behavior.

```
\section{Modeling individual behavior and decision making}
\label{sec:individual_behavior}
```

In a nutshell, models of individual decision making and behavior differ with regard to their assumptions about three crucial determinants of human choices: goals, restrictions and decision rules \citep{Hedstrom2005, Lindenberg2001, Lindenberg1990, Lindenberg1985}.
First, allthe models assume that individuals have motives or, goals or preferences. That is, agents rank goods or outcomes in terms of their desirability and seek to realize highly ranked outcomes. For instance, learning theories assume that actors evaluate the outcome of their choices and that satisficing decisions are reinforced. Other models, such as rational choice theory, make more complex assumptions about preference relations \citep{Neumann1944}. Another prominent but debated assumption is that motivesA prominent but debated assumption of many models is that preferences or goals are assumed to be stable over time. Stable preferences are included to prevent researchers from developing trivial explanations, as a theory that models a given change in behavior only based on changed motivespreferences does not have explanatory power. However, empirical research shows that preferences can change even in relatively short time frames \citep{Ackermann2016}. Furthermore, changingChanging individuals' goals or preferences is an important waymechanism to affect their behavior, e.g., through policies, making flexible preferences particularly interesting for Earth Systemsystem modelers.

Second, alldecision models make assumptions about restrictions and opportunities that constrain or help the agents to follow the motives orpursue their goals. For instance, each behavioral option comes with certain costs (e.g., money and time) and decision makers form more or less accurate beliefs about these costs and how likely they are to occur, depending on the information available to the agent.

Third, actorsmodels assume that agents apply some decision rule that translates their preferences and restrictions into a choice. Although decision rules differ very much in their complexity, they can be categorized into three types. First, there are decision rules that are forward looking. Rational choice theory, for

instance, assumes that individuals list all positive and negative future consequences of a decision and choose the optimal option. Alternatively, backwards looking approaches, such as classical reinforcement learning, assume that actors remember the satisfaction experienced when they chose a given behavior in the past and  choose a behavior with a high satisfaction again. Finally, there are sideward-looking decision rules, which assume that actors adopt the behavior of others, for instance because they imitate successful others \citep{Kandori1993}. Theories assume different degrees of context-dependency of rules and make different implicit assumptions about the underlying cognitive capabilities of agents.

In the remainder of this section, we describe in more detail three important approaches to individual decision making, pointing out typical assumptions about motives, restrictions and decision rules. ~~In Section~\ref{sec:discussion} we provide general guidelines for the choice of model assumptions.~~

**\subsection{Optimal decisions and utility theory in rational choice models}**
**\label{sec:rational_choice}**

\emph{Rational choice theory}, a standard model in many social sciences including economics and widely studied in mathematics, assumes that decision making is  goal-oriented : rational agents have \emph{preferences}} and choose the strategy whose expected outcome is most preferred, given some external \emph{constraints}}  and potentially based on their \emph{beliefs} \citep[represented by subjective probability distributions, see beliefs, preferences, constraints  model,][]{Gintis2009}.  It can either be used to represent actual behavior or serve as a normative benchmark for other theories of behavior.

How to judge the ``rationality'' of individual decisions is subject to ongoing debates.  \citet{Opp1999} distinguishes between  strong  rationality (``homo economicus''), assuming purely self-interested agents with unlimited cognitive capacities  possible actions and probabilities of consequences and weak rationality that makes less strong assumptions.  \citet{Rabin2002} distinguishes between standard and non-standard assumptions regarding preferences, beliefs and decision-making rules.  Before discussing non-optimal decision making in subsection \ref{sec:bounded_rationality},

we review here common assumptions on preferences
and beliefs.

Usually, agents are assumed to be mainly self-interested,
having fixed preferences regarding their personal consequences of
possible futures and
being indifferent to how a decision was taken and to consequences
for others.
Exceptions are procedural \citep{Hansson1996, Fehr1999}
and
other-regarding preferences \citep{Mueller2003, Fehr2003}.

Preferences can be modeled as binary
\emph{preference relations}, $x \ P_i \ y$, denoting that individual $i$ prefers situation or outcome $x$ to $y$.
Most authors assume that $P_i$ is complete (for every pair $(x, y)$ either $x P_i y$ or $y P_i x$) and transitive (if $x P_i y$ and $y P_i z$ then $x P_i z$), which allows representing the preferences with a \emph{utility function} $u_i$ if and only if$.\footnote{The utility function}. Utility functions thus specify how combinations of behavioral outcomes satisfy the preferences of the decision makers.~~.}
Some authors also allow incomplete or cyclic preferences \citep{Fishburn1968, Heitzig2012}.
In the land use context, $i$ could be a farmer, $x$ might denote
growing
some traditional crops generating a moderate profit,
and $y$ growing hybrid seeds for more profit
but making $i$ dependent on the seed supplier.
If $i$ considers independence valuable enough to make up for the lower profit,
$x \ P_i \ y$ would denote $i$'s preference of $x$ over $y$.

~~Utility functions are particularly useful in the context of \emph{decision making under uncertainty}\footnote{We note that some authors make the distinction between risk as unknown events with measurable probabilities (``known unknowns'') as opposed to (fundamental) uncertainty as such events without any knowledge about their probabilities \citep[``unknown unknowns'', cp.][]{Knight2006}. Although fundamental uncertainty may be important in human decision making, we only consider risk here because some forms of fundamental uncertainty cannot be represented in models.}.~~
In decision making under uncertainty, agents have to choose between different \emph{risky prospects} modeled as probability distributions

[revised manuscript text omitted]

Therefore, game theorists try to narrow down the likely strategy combinations by assuming additional forms of consistency and rationality \citep{Aumann2006} such as consistency over time (sequential and subgame perfect equilibria), and stability against small deviations \citep[stable equilibria][]{Foster1990}, or small random mistakes \citep[trembling-hand-perfect equilibria][]{Harsanyi1988}. After a plausible strategic equilibrium has been identified, it can be used in a simulation of the actual behavior resulting from these strategies over time, possibly including noise and mistakes.

As an example from the land-use context, consider two farmers living on the same road. They get their irrigation water from the same stream. A dispute over the use of water emerges. Both may react to the actions of the other in several turns. The upstream farmer located at the end of the road may increase or decrease her water use and/or pay compensation for using too much water to the other. The downstream farmer at the entrance of the road may demand compensation or block the road and thereby cut the access of the upstream farmer to other supplies. A complex game tree encodes which actions are feasible at which moment and what are the consequences on players' utilities. If it is possible to specify the information and options available to the players at each time point, then a classical game theoretical analysis allows determining the rational equilibrium strategies that the farmers would follow.

Classical game theory is widely applied to interactions in market settings in economics (see also Section~\ref{sec:macroeconomics}), but increasingly also in the social and political sciences to political and voting behavior in \emph{public and social choice theory} \citep[see, e.g.,][and Section~\ref{sec:social_welfare}]{Ordeshook1986, Mueller2003}. 
[revised manuscript text omitted]
 \citep{Holme2006}, epidemic spreading \citep{Gross2006}, the emergence of cooperation in social dilemmas  \citep{Perc2010} and the  coalition formation with social networks \citep{Auer2015}. Such adaptive network models exhibit complex and nonlinear  dynamics such as phase transitions \citep{Holme2006}, multi-stability \citep{Wiedermann2015}, oscillations in both agent states and network structure \citep{Gross2006}, and structural changes in network properties \citep{Schleussner2016}.

While adaptive networks have so far mostly been applied to networks of agents representing individuals, the framework can in principle be used to model co-evolutionary dynamics on various levels of social interaction as introduced in Table~\ref{tab:levels}.
For instance, global complex network structures such as financial risk networks between banks, trade networks between countries, transportation networks between cities and other communication, organizational and infrastructure networks can be modeled \citep{Currarini2016}. Furthermore, approaches such as multi-layer and hierarchical networks or networks of networks allow modeling the interactions between different levels of a system \citep{Boccaletti2014}.

As an illustration, consider a community of agents each harvesting a renewable resource, e.g., wood from a forest. The agents interact on a social network, imitating the harvesting effort of neighbors that harvest more and may drop links to neighbors that use another effort. The interaction of the resource dynamics with the network dynamics either leads to a convergence of harvest efforts or a segregation of the community into a group with a higher and a lower effort, depending on the model parameters \citep{Wiedermann2015, Barfuss2017}.

In the context of long time scales in the Earth system, the time evolution of social structures that determine interactions with the environment are particularly important. Adaptive networks offer a promising approach to modeling structural change of the internal connectivity of a complex system \citep{Lade2016}. For example, this could be applied to explore mechanisms behind transitions between centralized and decentralized infrastructure and organizational networks.

Table~\ref{tab:interaction} summarizes the different modeling approaches that focus on agent interactions in human decision making and behavior. These interactions occur between two or several agents. For including the effect of

these interactions into ESMs, their aggregate effects need to be taken into account as well. Therefore, we introduce in the next section approaches that allow to aggregate individual behavior and local interactions and to study the resulting macro-level dynamics.

```
%t
\begin{table*}[t]
\caption{Summary table for agent interactions.}
\label{tab:interaction}
\begin{tabular}{L{3cm}p{4cm}p{4cm}p{4cm}}
\tophline
Approaches and frameworks & Key considerations & Strengths & Limitations \\
\middlehline

[revised manuscript text omitted]

\bottomhline
\end{tabular}
\belowtable{} % Table Footnotes
\end{table*}

%
================================================================================
%
================================================================================

\section{Discussion}
\label{sec:discussion}

In the previous three sections, we reviewed different showed that there is a
diversity of approaches to model _individual human decision making and behavior,
to describe interactions between agents and to aggregate these processes. We
illustrated them with examples from the land-use context and discussed their
potential application in Earth system modeling with the aim of modeling complex
feedback dynamics between natural and social components.

One intentionThe discussion of this review is to draw the attentionstrengths and
limitations of the reader to the differentmodeling approaches showed possible
underlying assumptions and connections to theories about human decision making
```

[revised manuscript text omitted]

```
%
================================================================================
%
================================================================================

%\authorcontribution{TEXT}

\competinginterests{The authors declare that they have no conflict of interest.}

%\disclaimer{TEXT}
```

```
\begin{acknowledgements}
```
The idea for this paper emerged during discussions at the LOOPS workshop 2014. We thank all the participants for their contributions. Furthermore, we thank the referees and the editor for helpful comments that improved the manuscript. We thank Rainer Hegselmann, Silvana Tiedemann, Tim Kittel, Wolfram Barfuss, Nicola Botta and Felix Kersting for their comments on previous versions of the paper. FMH acknowledges funding by the DFG (IRTG 1740/TRP 2011/50151-0). MS acknowledges funding by the European Research Council under the European Union's Seventh Framework Programme (FP/2007-2013)/ERC grant agreement no. 283950 SES-LINK and a core grant to the Stockholm Resilience Centre by Mistra. JFD thanks the Stordalen foundation (via the Planetary Boundary Research Network PB.net), the Earth Doc network and the Leibniz Association (project DOMINOES) for financial support.
```
\end{acknowledgements}
```

```
%% REFERENCES

\bibliographystyle{copernicus}
\bibliography{Literature_ESD_Review.bib}

\end{document}
```